# Ulk4 promotes Shh signaling by regulating Stk36 ciliary localization and Gli2 phosphorylation

**Mengmeng Zhou[1], Yuhong Han[1], Jin Jiang[1,2]\***

[1]Department of Molecular Biology, University of Texas Southwestern Medical Center, Dallas, United States; [2]Department of Pharmacology, University of Texas Southwestern Medical Center, Dallas, United States

**Abstract** The Hedgehog (Hh) family of secreted proteins governs embryonic development and adult tissue homeostasis through the Gli family of transcription factors. Gli is thought to be activated at the tip of primary cilium, but the underlying mechanism has remained poorly understood. Here, we show that *U*nc-51-*l*ike *k*inase 4 (Ulk4), a pseudokinase and a member of the Ulk kinase family, acts in conjunction with another Ulk family member Stk36 to promote Gli2 phosphorylation and Hh pathway activation. Ulk4 interacts with Stk36 through its N-terminal region containing the pseudokinase domain and with Gli2 via its regulatory domain to bridge the kinase and substrate. Although dispensable for Hh-induced Stk36 kinase activation, Ulk4 is essential for Stk36 ciliary tip localization, Gli2 phosphorylation, and activation. In response to Hh, both Ulk4 and Stk36 colocalize with Gli2 at ciliary tip, and Ulk4 and Stk36 depend on each other for their ciliary tip accumulation. We further show that ciliary localization of Ulk4 depends on Stk36 kinase activity and phosphorylation of Ulk4 on Thr1023, and that ciliary tip accumulation of Ulk4 is essential for its function in the Hh pathway. Taken together, our results suggest that Ulk4 regulates Hh signaling by promoting Stk36-mediated Gli2 phosphorylation and activation at ciliary tip.

**\*For correspondence:**
jin.jiang@utsouthwestern.edu

**Competing interest:** The authors declare that no competing interests exist.

## eLife assessment

This **fundamental** study substantially advances our understanding of how the pseudokinase ULK4 interacts with an active member of the same kinase subfamily (STK36) to promote GLI phosphorylation and Hedgehog pathway activation. The evidence supporting the proposed mechanism is **compelling**, with rigorous biochemical assays and state-of-the-art cell-based imaging techniques. The work will be of broad interest to cell biologists and biochemists.

## Introduction

Pseudokinases, which comprise ~10% of the mammalian kinome, play essential biological functions, yet the mechanism of action for many pseudokinases has remained unknown (*Jacobsen and Murphy, 2017*). One such pseudokinase is *U*nc-51-*l*ike *k*inase 4 (Ulk4), which is a member of Ulk kinase family found in most eukaryotes (*Luo et al., 2022*). The Ulk family kinases, which include Ulk1 (Atg1 in yeast) and Ulk2 involved in autophagy initiation, contain a conserved kinase domain in their N-terminal regions and a variable regulatory domain in their C-terminal regions (*Luo et al., 2022*). However, unlike other Ulk family members, Ulk4 contains a pseudokinase domain at its N-terminus and is predicted to be catalytically inactive (*Preuss et al., 2020*). Ulk4 plays a key role in neurogenesis in mice by regulating neural stem cell pool and neurite branching morphogenesis (*Lang et al., 2014*; *Liu et al., 2016a*). In addition, Ulk4 regulates the integrity of white matter by promoting myelination

(*Liu et al., 2018b*). Recent studies have implicated Ulk4 as a risk factor for neuropsychiatric disorders, including schizophrenia (*Lang et al., 2014*; *Lang et al., 2016*; *Liu et al., 2018a*; *Luo et al., 2022*). However, the signaling pathways in which Ulk4 is involved and the substrates it regulates have remained unknown. A recent proteome study has revealed that Ulk4 interacts with Stk36 in addition to many other proteins (*Preuss et al., 2020*). Stk36 is also a member of the Ulk family and mammalian homolog of *Drosophila* Fused (Fu), a Ser/Thr kinase implicated in Hedgehog (Hh) signaling (*Murone et al., 2000*; *Wilson et al., 2009*; *Han et al., 2019*). A recent study identified Ulk4 as a genetic modifier of the holoprosencephaly phenotype caused by impaired Shh pathway activity (*Mecklenburg et al., 2021*). In addition, this study showed that Ulk4 was localized to the primary cilia and that Ulk4 knockdown affected Shh induced *Gli-luc* reporter gene expression and Smoothened (Smo) ciliary localization (*Mecklenburg et al., 2021*). These observations have raised an interesting possibility that Ulk4 may participate in Hh signal transduction.

The Hh family of secreted proteins governs embryonic development and adult tissue homeostasis, and deregulation of Hh signaling activity has been implicated in a myriad of human disorders, including birth defect and cancer (*Villavicencio et al., 2000*; *Jiang and Hui, 2008*; *Briscoe and Thérond, 2013*; *Jiang, 2022*). Hh exerts its biological influence via an evolutionarily conserved signaling cascade that culminates in the activation of Gli family of Zn-finger transcription factors. The core Hh signal reception system consists of two multi-span transmembrane proteins Patched1 (Ptch1) and Smoothened (Smo) that transduce vertebrate Hh signal at primary cilia (*Goetz and Anderson, 2010*). In quiescent cells, Ptch1 localizes to primary cilia where Smo is barely detectable (*Corbit et al., 2005*; *Rohatgi et al., 2007*); the full-length Gli (Gli$^F$) undergoes ubiquitin/proteasome-mediated proteolytic processing to generate a truncated transcriptional repressor form (Gli$^R$) (*Wang et al., 2000*; *Jiang, 2002*; *Tempé et al., 2006*; *Wang and Li, 2006*; *Chen and Jiang, 2013*). Hh inhibits Ptch1 activity and promotes its ciliary exit, allowing Smo and Gli proteins to accumulate in the cilia to transduce the Hh signal (*Corbit et al., 2005*; *Haycraft et al., 2005*; *Rohatgi et al., 2007*; *Chen et al., 2009*; *Tukachinsky et al., 2010*; *Chen et al., 2011*). Activated Smo blocks the proteolytic processing of Gli into Gli$^R$ and converts Gli$^F$ to an active form (Gli$^A$) that enters the nucleus to turn on the expression of Hh target genes (*Wang et al., 2000*; *Kim et al., 2009*; *Tukachinsky et al., 2010*; *Han et al., 2019*; *Arveseth et al., 2021*).

Although it is generally thought that Smo converts Gli$^F$ into Gli$^A$ at ciliary tip (*Goetz and Anderson, 2010*), the underlying mechanism has remained an enigma. Our recent studies have demonstrated that two Ulk family kinases Ulk3 and Stk36 play a role in converting Gli$^F$ into Gli$^A$ by directly phosphorylating Gli2 at multiple sites in mammalian cells (*Han et al., 2019*; *Zhou et al., 2022*; *Zhou and Jiang, 2022*). Here, we provide evidence that Ulk4 acts in conjunction with Stk36 to promote Gli2 phosphorylation and Hh pathway activation. We show that Ulk4 interacts with Stk36 through its N-terminal region containing the pseudokinase domain and with Gli2 via its C-terminal regulatory domain to bring Stk36 and Gli2 into proximity. We find that Ulk4 is not required for Stk36 kinase activation by Shh but is essential for Stk36 ciliary localization and Gli2 phosphorylation. We show that, in response to Shh stimulation, both Ulk4 and Stk36 colocalize with Gli2 at the tips of primary cilia, and that Ulk4 and Stk36 depend on each other for their ciliary tip accumulation. We further show that ciliary tip localization of Ulk4 depends on Stk36-mediated phosphorylation of Ulk4 on Thr1023, and that ciliary tip localization of Ulk4 is essential for its function in Hh signal transduction.

## Results

### Ulk4 acts in conjunction with Stk36 to promote Gli2 phosphorylation and Hh pathway activation

To determine whether Ulk4 is required for Hh pathway activation, we depleted Ulk4 by RNA interference (RNAi) in NIH3T3 cells, a well-established system for studying Hh signaling. We found that Ulk4 depletion significantly decreased Shh-induced expression of *Gli1* and *Ptch1* determined by RT-qPCR, which was rescued by the expression of human Ulk4 (hUlk4; *Figure 1A and B*, *Figure 1—figure supplement 1*). The requirement of Ulk4 in Shh-induced expression of *Gli1* and *Ptch1* was confirmed in mouse embryonic fibroblast (MEF) cells (*Figure 1—figure supplement 2*).

Our previous study revealed that Stk36 and Ulk3, which are Ulk family kinases, act in parallel to promote Gli2 phosphorylation and Hh pathway activation (*Han et al., 2019*). To determine the relationship between Ulk4 and Ulk3/Stk36, we carried out double RNAi experiments. We found that

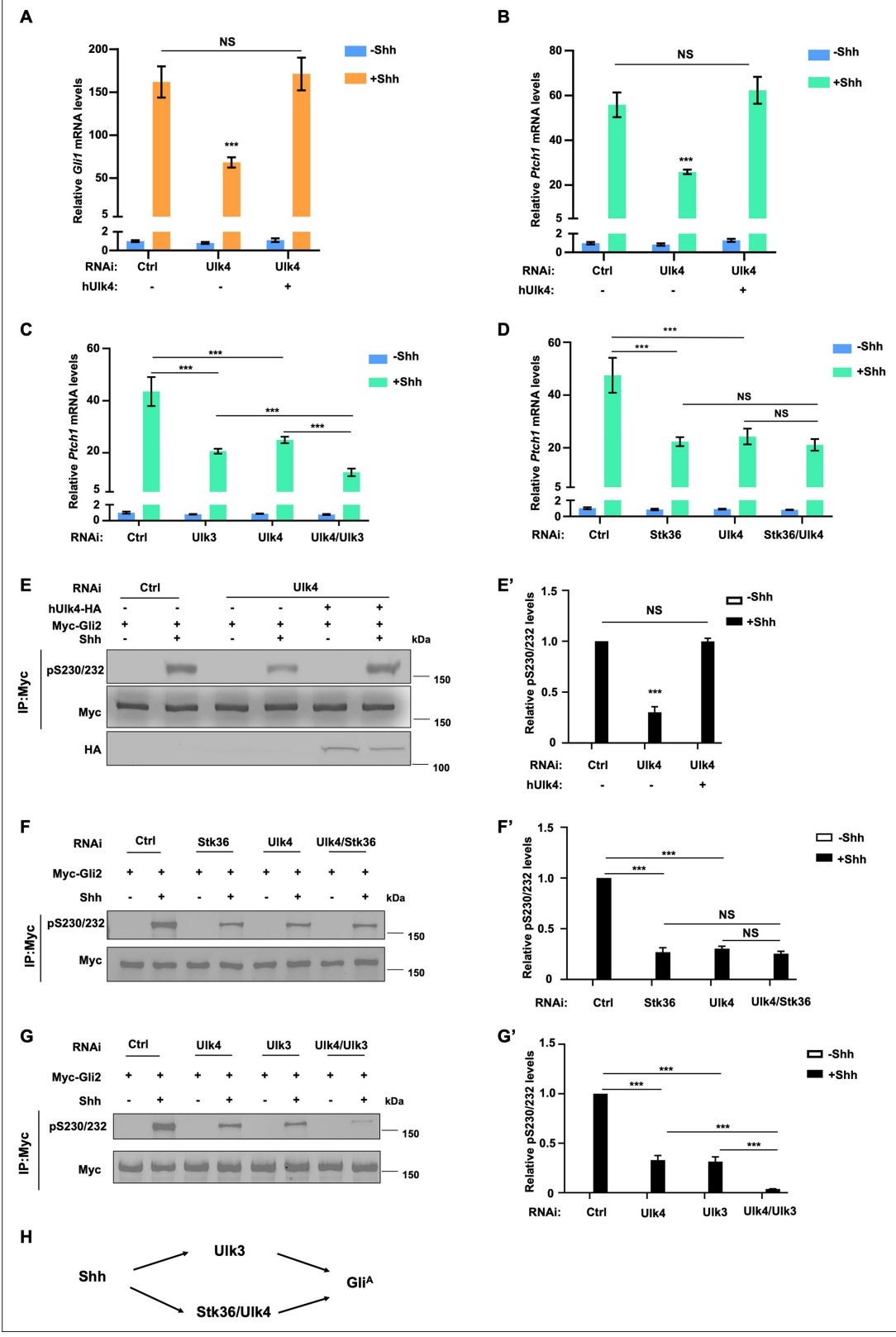

**Figure 1.** Ulk4 acts in conjunction with Stk36 to promote Shh-mediated Gli activation. (**A, B**) Relative *Gli1* (**A**) and *Ptch1* (**B**) mRNA levels measured by RT-qPCR in NIH3T3 cells expressing the indicated shRNA and a human Ulk4 (hUlk4) lentiviral construct and treated with or without Shh-N. Of note, hUlk4 is insensitive to RNAi, which targets mouse Ulk4. (**C, D**) Relative *Ptch1* mRNA levels measured by RT-qPCR in NIH3T3 cells treated with the indicated RNAi and with or without Shh-N. (**E, E'**) Western blot analysis (**E**) and quantification (**E'**) of Myc-Gli2 phosphorylation in NIH3T3 cells expressing the

*Figure 1 continued on next page*

*Figure 1 continued*

indicated shRNA and hUlk4 lentiviral construct and treated with or without Shh-N. (**F–G′**) Western blot analysis (**F, G**) and quantification (**F′, G′**) of Myc-Gli2 phosphorylation in NIH3T3 cells with the indicated RNAi in the presence or absence of Shh-N treatment. (**H**) Schematic diagram showing the functional relationship among Ulk4, Stk36, and Ulk3 in Shh-induced Gli activation. The cells (**A–G**) were first starved in serum-free medium for 12 hr, then cultured in the same medium with or without Shh-N fragment for another 12 hr before they are subjected to RNA preparation (**A–D**) or western blot analysis (**E–G**). Data in (**A–D**) are mean ± SD from three independent experiments. Data in (**E′, F′, G′**) are mean ± SD from two independent experiments. ***p<0.001 (Student's *t*-test). NS, not significant.

The online version of this article includes the following source data and figure supplement(s) for figure 1:

**Source data 1.** Source data for western blots in *Figure 1E, F, and G*.

**Figure supplement 1.** The knockdown efficiency of Ulk4, Ulk3, and Stk36 in NIH3T3 cells.

**Figure supplement 2.** Ulk4 acts in conjunction with Stk36 to regulate Shh-induced expression of *Gli1* and *Ptch1* in mouse embryonic fibroblasts (MEFs).

double knockdown (KD) of Ulk4 and Ulk3 further decreased Shh target gene expression compared with single KD of either Ulk4 or Ulk3, whereas double KD of Ulk4 and Stk36 did not have an additive effect (*Figure 1C and D*, *Figure 1—figure supplement 2*). These results indicated that Ulk4 acts in the same pathway with Stk36 but in parallel with Ulk3 to promote Hh signaling.

Our previous study demonstrated that Ulk3 and Stk36 phosphorylate Gli2 at multiple sites including S230/232 to promote Gli2 activation (*Han et al., 2019*; *Zhou et al., 2022*). We found that Ulk4 KD reduced Gli2 phosphorylation on S230/232 stimulated by Shh in NIH3T3 cells, which was determined by western blot analysis using a phospho-specific antibody that recognizes pS230/232 (*Han et al., 2019*; *Han and Jiang, 2021*), and that expression of hUlk4 restored Gli2 phosphorylation on S230/232 in Ulk4-depleted cells (*Figure 1E and E′*). Consistent with Ulk4 acting in a liner pathway with Stk36 but in parallel with Ulk3, double KD of Ulk4 and Stk36 decreased S230/232 phosphorylation similarly to single KD of either Ulk4 or Stk36, whereas double KD of Ulk3 and Ulk4 further decreased Shh-induced S230/232 phosphorylation compared with single KD of Ulk3 or Ulk4 (*Figure 1F–G′*). Taken together, these results demonstrated that Ulk4 and Stk36 act in a linear pathway but in parallel with Ulk3 to participate in Hh transduction by facilitating Gli2 phosphorylation and activation (*Figure 1H*).

## Ulk4 forms a complex with Stk36 and Gli2

A previous study using proximity-dependent biotin identification (BioID) identified Stk36 as a proximity interactor (*Preuss et al., 2020*). To verify that Stk36 formed a complex with Ulk4, we carried out co-immunoprecipitation (Co-IP) experiments and found that a C-terminal HA-tagged Ulk4 (hUlk4-HA) interacted with an N-terminal CFP-tagged Stk36 (CFP-Stk36) in HEK293T cells (*Figure 2A*). To explore whether Ulk4 interacted with Gli2 and Sufu, a Gli binding partner and Hh pathway inhibitor (*Dunaeva et al., 2003*; *Cooper et al., 2005*; *Han et al., 2015*), we co-expressed hUlk4-HA, with a Myc-tagged Gli2 (Myc-Gli2) and a Flag (Fg)-tagged Sufu 0 in HEK293T cells. Co-IP experiments showed that both Myc-Gli2 and Fg-Sufu were pulled down by Ulk4-HA (*Figure 2B*), suggesting that Ulk4 could form a complex with Stk36 and Gli2-Sufu.

Ulk4 contains an N-terminal pseudokinase domain and the C-terminal regulatory domain with armadillo (Arm) repeats (*Figure 2C*; *Preuss et al., 2020*). We generated two truncated forms of Ulk4: hUlk4N (aa 1–450) that contains the pseudokinase domain and hUlk4C (451–1275) that contains the C-terminal regulatory domain. Co-IP experiments showed that hUlk4N-HA interacted with Fg-Stk36 but not with Myc-Gli2/Fg-Sufu, whereas hUlk4C-HA formed a complex with Myc-Gli2/Fg-Sufu but not with Fg-Stk36 (*Figure 2D and E*), suggesting that Ulk4 interacted with Stk36 and Gli2-Sufu through its N-terminal pseudokinase domain and C-terminal regulatory domain, respectively. Consistent with our observation, previous BioID experiments using BirA*-Ulk4 pseudokinase domain also identified Stk36 as a proximity binding partner of the Ulk4 pseudokinase domain (*Preuss et al., 2020*).

## Ulk4 is dispensable for Shh-induced Stk36 kinase activation

Pseudokinases could regulate the enzymatic activity of interacting kinases, as has been shown for LKB1 kinase activation by its binding to the pseudokinases STRADα and STARADβ (*Baas et al., 2003*). To determine whether Ulk4 regulates Stk36 enzymatic activity, we carried out an *in vitro* kinase assay using GST-Gli2N, which contains the Ulk3/Stk36 phosphorylation site S230, as substrate and either Fg-Stk36 or Fg-Stk36/hUlk4-HA complex immunopurified from HEK293T cells as kinase. GST-Gli2N

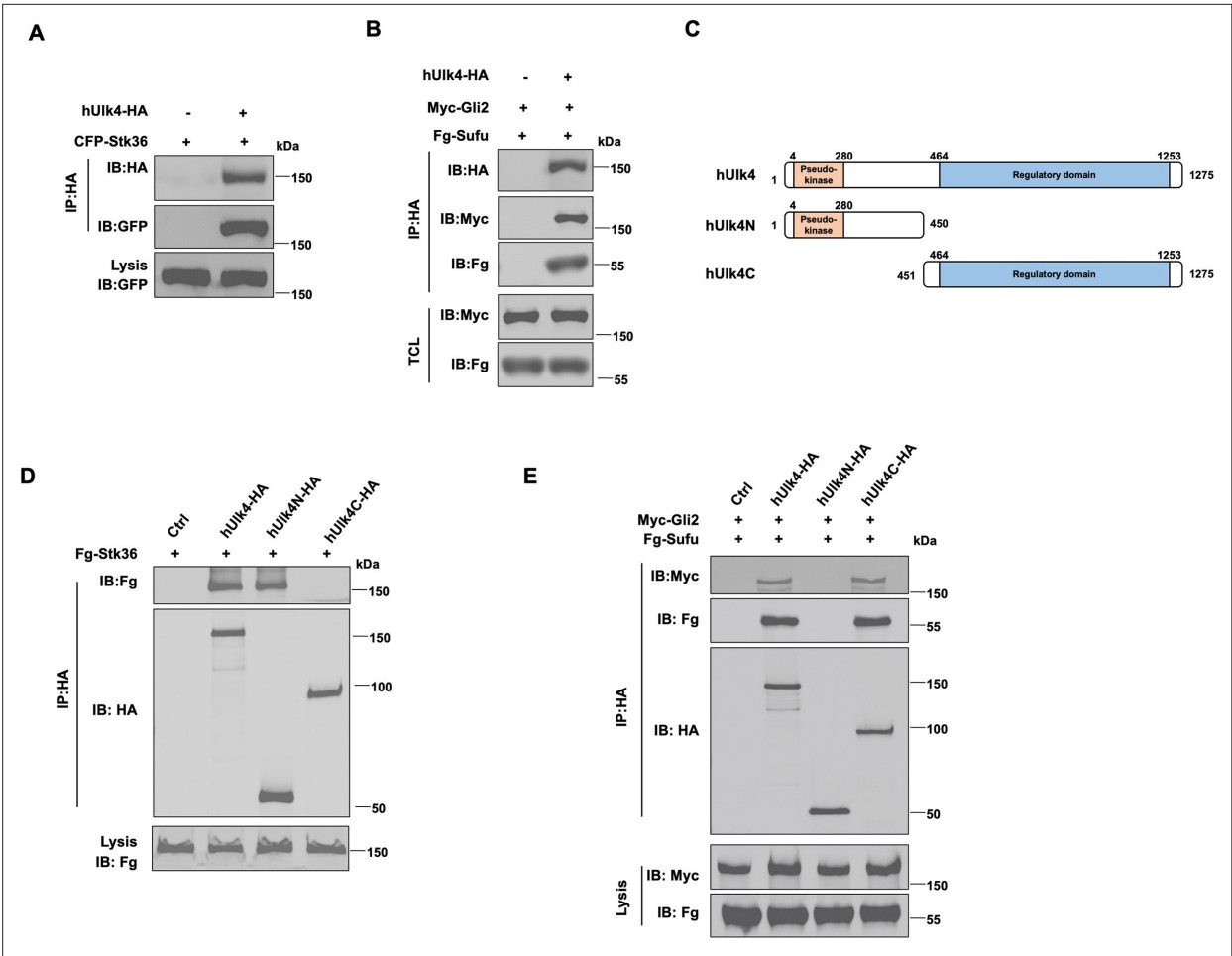

**Figure 2.** Ulk4 forms a complex with Stk36 and Gli2. (**A**) Ulk4 forms a complex with Stk36. HEK293T cells were transfected with hUlk4-HA and CFP-Stk36 constructs, followed by co-immunoprecipitation (Co-IP) and western blot analyses with the indicated antibodies. (**B**) Ulk4 forms a complex with Gli2 and Sufu. HEK293T cells were transfected with hUlk4-HA, Myc-Gli2, and Fg-Sufu constructs, followed by Co-IP and western blot analyses with the indicated antibodies. (**C**) Ulk4 domain structure and deletion mutants used for Co-IP experiments. (**D, E**) Ulk4 interacted with Stk36 through its N-terminal domain (**D**) and with Myc-Gli2/Fg-Sufu through its C-terminal region (**E**). HEK293T cells were transfected with the indicated hUlk4, Fg-Stk36, Myc-Gli2, and Fg-Sufu constructs, followed by Co-IP and western blot analyses.

The online version of this article includes the following source data for figure 2:

**Source data 1.** Source data for western blots in **Figure 2A, B, D, and E**.

containing S230A substitution was used as a negative control. As shown in **Figure 3A**, Stk36 and Stk36/Ulk4 phosphorylated GST-Gli2N at similar levels and the phosphorylation was abolished by the S230A mutation. This result suggests binding of Ulk4 to Stk36 does not modulate Stk36 kinase activity.

We next determined whether Ulk4 is required for Stk36 kinase activation by Shh. In *Drosophila*, Hh activates Fu by promoting its kinase domain dimerization and trans-autophosphorylation on multiple S/T residues in its activation loop (AL), including T151, T154, T158, and S159 (**Shi et al., 2011**; **Zhang et al., 2011**; **Zhou and Kalderon, 2011**). Interestingly, the AL phosphorylation sites as well as the neighboring residues are highly conserved between Fu and Stk36 (**Figure 3B**; **Shi et al., 2011**), raising the possibility that Stk36 could be activated by Shh through a similar mechanism. Indeed, we found that Shh induced phosphorylation of Stk36 on T158/S159 in NIH-3T3 cells as determined by a phospho-specific antibody that recognized pT158/pS159 (**Figure 3C**; **Shi et al., 2011**). To determine whether Shh-induced T158/S159 phosphorylation depends on Stk36 kinase activity, we introduced either wild type Stk36 (CFP-Stk36$^{WT}$) or its kinase dead form (CFP-Stk36$^{KR}$) into NIH3T3 cells with endogenous Stk36 and Ulk3 deleted (*Ulk3$^{KO}$ Stk36$^{KO}$*) by CRISPR (**Han et al., 2019**). As

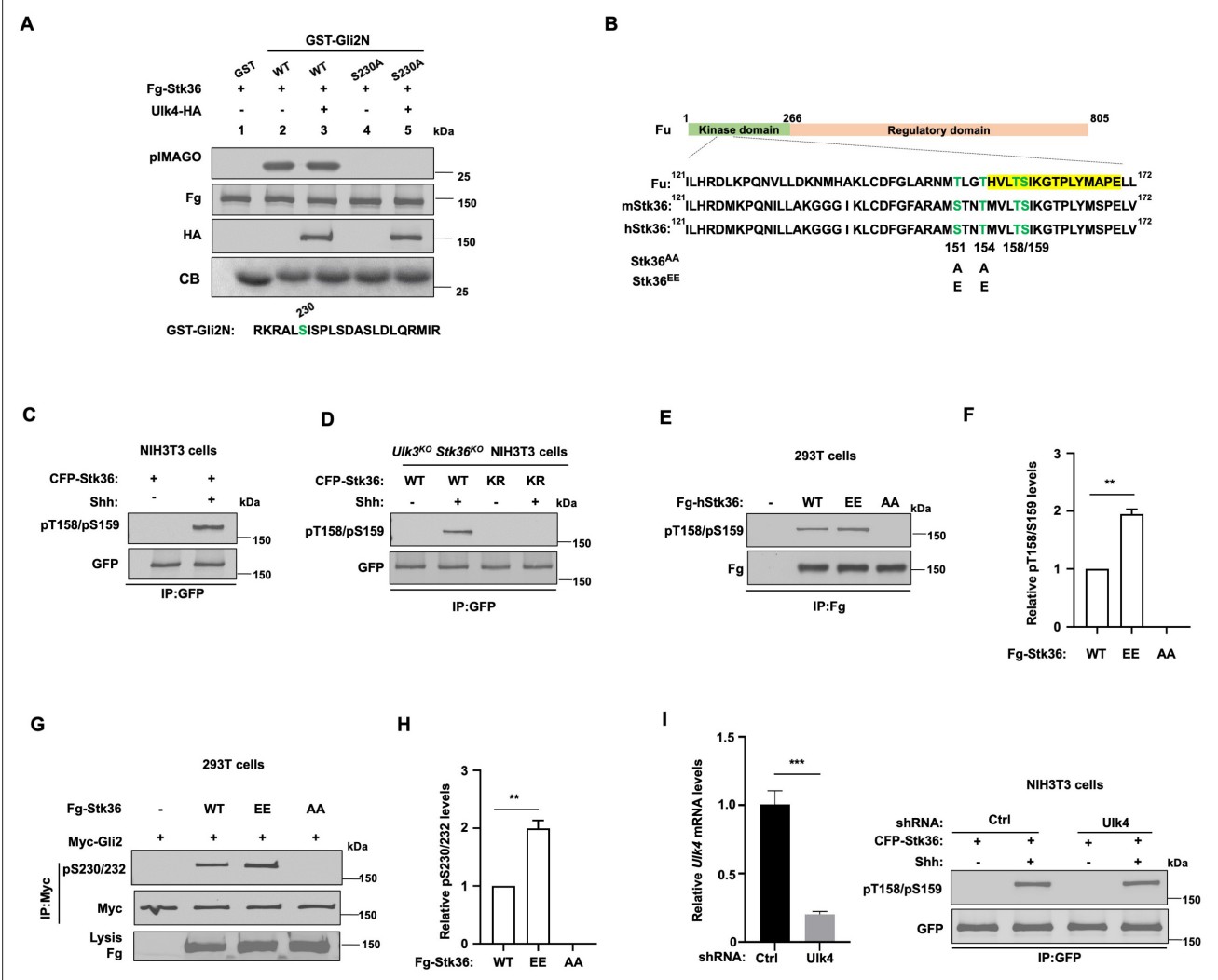

**Figure 3.** Ulk4 is dispensable for Stk36 kinase activation. (**A**) *In vitro* kinase assay using the immunopurified Fg-Stk36 or Fg-Stk36/hUlk4-HA as kinase and GST-Gli2N[WT] or GST-Gli2N[S230A] as substrate. pIMAGO was used to detect GST-Gli2N phosphorylation. The levels of Fg-Stk36/hUlk4-HA and GST/GST-Gli2N were analyzed by western blot and Coomassie blue (CB) staining, respectively. (**B**) The schematic diagram showing the sequence alignment of the activation segments of *Drosophila* Fu, mouse Stk36 (mStk36), and human Stk (hStk36). The conserved Ser/Thr residues in the activation loop are color-coded in green. The residues used as antigen for developing pT158/pS159 are colored in yellow. Amino acid substitutions for Stk36 variants are indicated. (**C**) Shh induced Stk36 phosphorylation on T158/S159. NIH3T3 cells expressing CFP-Stk36 lentiviral construct were treated with or without Shh-N, followed by. IP with a GFP antibody and western blot analysis using an antibody against the phosphorylated T158/S159 (pT158/pS159). (**D**) Shh induced Stk36 phosphorylation on T158/S159 depending on Stk36 kinase activity. Ulk3 and Stk36 double knockout NIH3T3 cells expressing CFP-Stk36[WT] or CFP-Stk36[KR] (kinase dead) lentiviral construct were treated with or without Shh-N, followed by IP with a GFP antibody and western blot analysis with the pT158/pS159 antibody. (**E, F**) Western blot analysis (**E**) and quantification (**F**) of T158/S159 phosphorylation in Fg-Stk36[WT], Fg-Stk36[EE], or Fg-Stk36[AA]. HEK293T cells were transfected with the indicated Fg-Stk36 constructs, followed by IP with anti-Flag antibody and western blot analysis with the indicated antibodies. (**G, H**) Western blot analysis (**G**) and quantification (**H**) of Myc-Gli2 phosphorylation on S230/S232 by the wild type (WT) or indicated Stk36 variants in HEK293T cells. (**I**) Ulk4 is not required for Shh-induced Stk36 kinase activation. NIH3T3 cells expressing the indicated shRNA and CFP-Stk36 lentiviral construct were treated with or without Shh-N, followed by IP with a GFP antibody and western blot analysis with the pT158/pS159 antibody (right). The knockdown efficiency of Ulk4 was determined by RT-qPCR (left). Data are mean ± SD from three independent experiments. **p<0.01, ***p<0.001 (Student's *t*-test).

The online version of this article includes the following source data for figure 3:

**Source data 1.** Source data for western blots in *Figure 3A, C–E, G, and I*.

shown in *Figure 3D*, Shh induced phosphorylation of CFP-Stk36$^{WT}$ but not CFP-Stk36$^{KR}$ on T158/S159, suggesting that Shh promotes the phosphorylation of Stk36 AL via its kinase activity, likely through trans-autophosphorylation as are the cases for other kinases. Hence, phosphorylation of T158/S159 can be used as a readout for Stk36 kinase activation.

To determine whether AL phosphorylation promotes Stk36 kinase activation, we converted both S151 and T154 to E (EE) to mimic phosphorylation or to A (AA) to block phosphorylation on these sites (*Figure 3B*). We found that the phospho-mimetic mutation (Stk36$^{EE}$) increased, whereas phospho-deficient mutation (Stk36$^{AA}$) abolished Stk36 kinase activity as determined by both Stk36 T158/S159 phosphorylation and Gli2 S230/S232 phosphorylation (*Figure 3E–H*), which is similar to what we have observed previously for Fu (*Shi et al., 2011*), suggesting that Hh stimulates kinase activation of Stk36 and Fu through a similar mechanism.

To determine whether Ulk4 is required for Shh-induced Stk36 kinase activation, we employed RNAi to deplete Ulk4 from NIH3T3 cells that express CFP-Stk36. As shown in *Figure 3I*, depletion of Ulk4 did not affect Shh-induced phosphorylation of CFP-Stk36 on T158/S159, suggesting that Ulk4 is not required for Stk36 kinase activation in response to Shh stimulation.

## Ulk4 mediates the interaction between Stk36 and Gli2

Another possible mechanism for Ulk4 to influence Gli2 phosphorylation by Stk36 is to function as a scaffold that brings Stk36 and Gli2 into the same protein complex. Indeed, we found that Ulk4 depletion diminished the association between exogenously expressed Stk36 and Gli2 in HEK293T cells as determined by Co-IP experiments (*Figure 4A*). By domain mapping, we found that the kinase domain of Stk36 (Stk36N: aa1-260) but not its regulatory domain (Stk36C: aa261-1351) interacted with Ulk4 (*Figure 4B and C*). Consistent with Ulk4 mediating the interaction between Stk36 and Gli2, we found that Stk36N formed a complex with Gli2 more effectively than Stk36C (*Figure 4D*).

We next determined whether Ulk4 mediated Stk36/Gli2 interaction in NIH3T3 cells and whether this was regulated by Shh. NIH3T3 cells were infected with lentiviruses expressing low levels of a Flag-tagged Stk36 (Fg-Stk36) and Myc-Gli2 and treated with or without Shh. Co-IP experiments showed that Fg-Stk36 and Myc-Gli2 formed a complex regardless of Shh treatment (*Figure 4E*; lands 1–2). The interaction between Fg-Stk36 and Myc-Gli2 was diminished by Ulk4 RNAi (*Figure 4E*; lands 3–4). Reintroducing full-length hUlk4 (WT) but not hUlk4N or hUlk4C restored Fg-Stk36/Myc-Gli2 association in Ulk4-depleted cells (*Figure 4E*; lands 5–10). Taken together, these results suggest that Ulk4 functions as a molecular scaffold to bring Stk36 and Gli2 into the same complex with its N-terminal region binding to Stk36 and its regulatory domain interacting with Gli2 (*Figure 4F*).

## Shh stimulates ciliary tip accumulation of both Ulk4 and Stk36

In response to Shh, Gli2 and Sufu accumulated at the tip of primary cilium in mammal cells (*Haycraft et al., 2005*; *Chen et al., 2009*; *Tukachinsky et al., 2010*), leading to the general view that Gli2 is converted to Gli2$^A$ at the ciliary tip (*Goetz and Anderson, 2010*; *Jiang, 2022*). Indeed, our previous study showed that Shh-stimulated Gli2 phosphorylation by Stk36/Ulk3 depended on ciliary localization of Gli2 (*Han et al., 2019*). These observations imply that Stk36/Ulk3 may translocate to the primary cilium to phosphorylate Gli2 in response to Shh. To test this hypothesis, we treated NIH3T3 cells stably expressing an N-terminal Myc-tagged Stk36 (Myc-Stk36) with or without Shh and examined ciliary localization of Stk36 by immunostaining with an anti-Myc antibody. Ciliary localization of endogenous Gli2 was examined by immunostaining with an anti-Gli2 antibody. As shown in *Figure 5A and B* and *Figure 5—figure supplement 1A1–B2*, Myc-Stk36 (green signals) and Gli2 (blue signals) were barely detectable in primary cilia in quiescent cells; however, Shh treatment induced ciliary tip localization of both Myc-Stk36 and Gli2.

To determine whether Shh induced ciliary localization of Ulk4, NIH3T3 cells stably expressing hUlk4-HA were treated with or without Shh, followed by immunostaining with anti-HA and anti-Gli2 antibodies. Like Stk36, hUlk4-HA was barely detectable in primary cilia in the absence of Shh but accumulated to the tips of primary cilia and colocalized with Gli2 after Shh stimulation (*Figure 5C and D*, *Figure 5—figure supplement 1C1–D2*). We also examined NIH3T3 cells simultaneously expressing Myc-Stk36 and hUlk4-HA and found that Shh induced the ciliary tip co-localization of Ulk4 and Stk36 (*Figure 5E and F*, *Figure 5—figure supplement 1E1–F2*). We confirmed these results in MEF cells

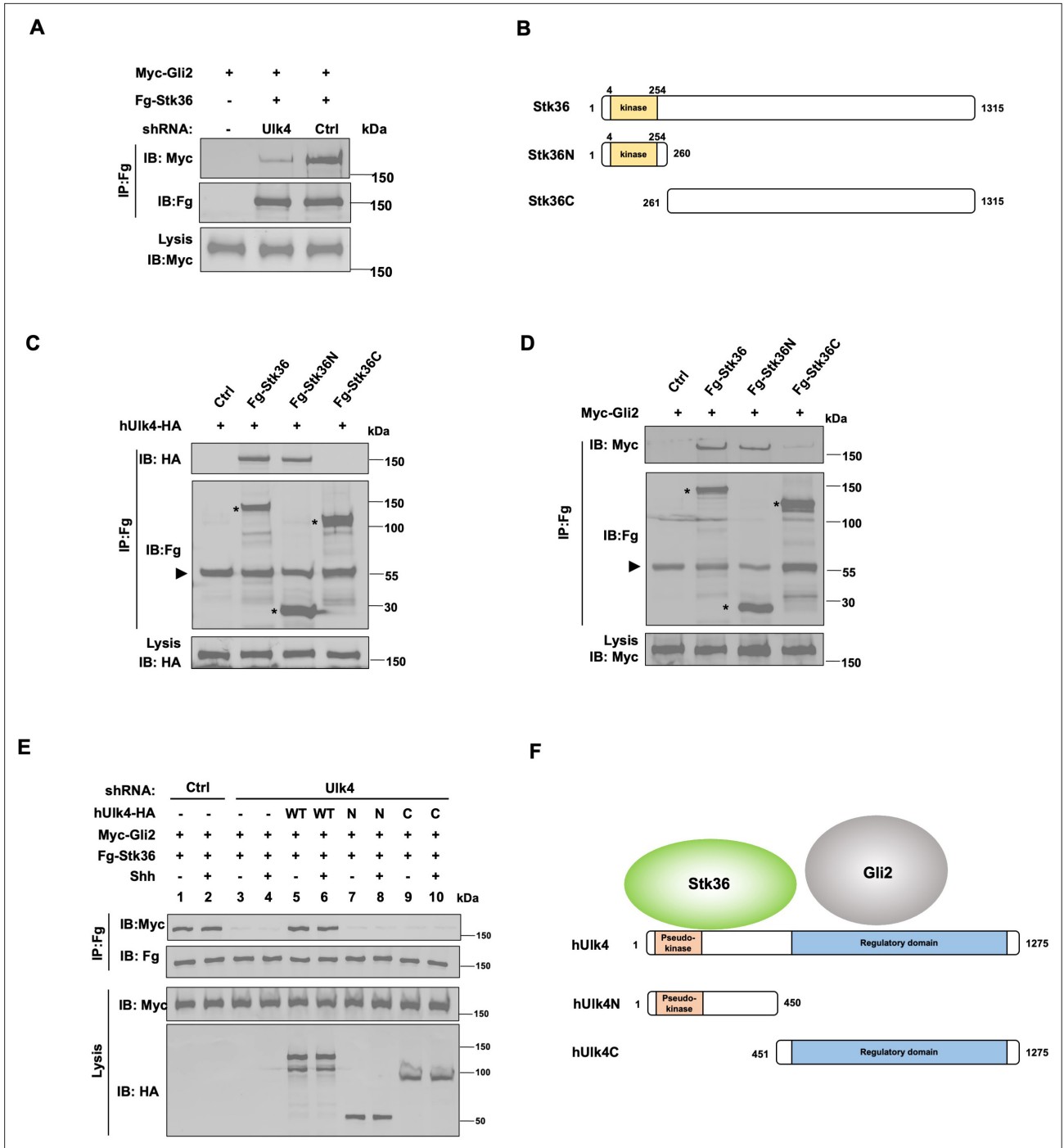

**Figure 4.** Ulk4 mediates the interaction between Stk36 and Gli2. (**A**) Depletion of Ulk4 diminished the association between Stk36 and Gli2. HEK293T cells were transfected with the indicated shRNA and Fg-Stk36 and Myc-Gli2 constructs, followed by co-immunoprecipitation (Co-IP) and western blot analysis with the indicated antibodies. (**B**) Domain structure and deletion mutants of human Stk36 used for the Co-IP experiments. (**C, D**) Stk36 binds to Ulk4 and Gli2 through its N-terminal kinase domain. HEK293T cells were transfected with Ulk4-HA (**C**) or Myc-Gli2 (**D**) and the indicated Fg-Stk36 constructs, followed by Co-IP and western blot analyses with the indicated antibodies. Black triangles indicate the heavy chain of IgG, and asterisks indicate the protein bands produced by individual Stk36 constructs. (**E**) Both N- and C-terminal domains of Ulk4 are required for mediating Stk36/Gli2 association. NIH3T3 cell lines stably expressing control (Ctrl) or Ulk4 shRNA were infected with lentiviruses expressing the indicated hUlk4 constructs (diagrams shown on **F**, right), Fg-Stk36 and Myc-Gli2, and treated with or without the Shh-N, followed by Co-IP and western blot analysis with the indicated antibodies. (**F**) Schematic diagram showing the interaction relationship between Ulk4 and Stk36, Gli2.

The online version of this article includes the following source data for figure 4:

**Source data 1.** Source data for western blots in *Figure 4A and C–E*.

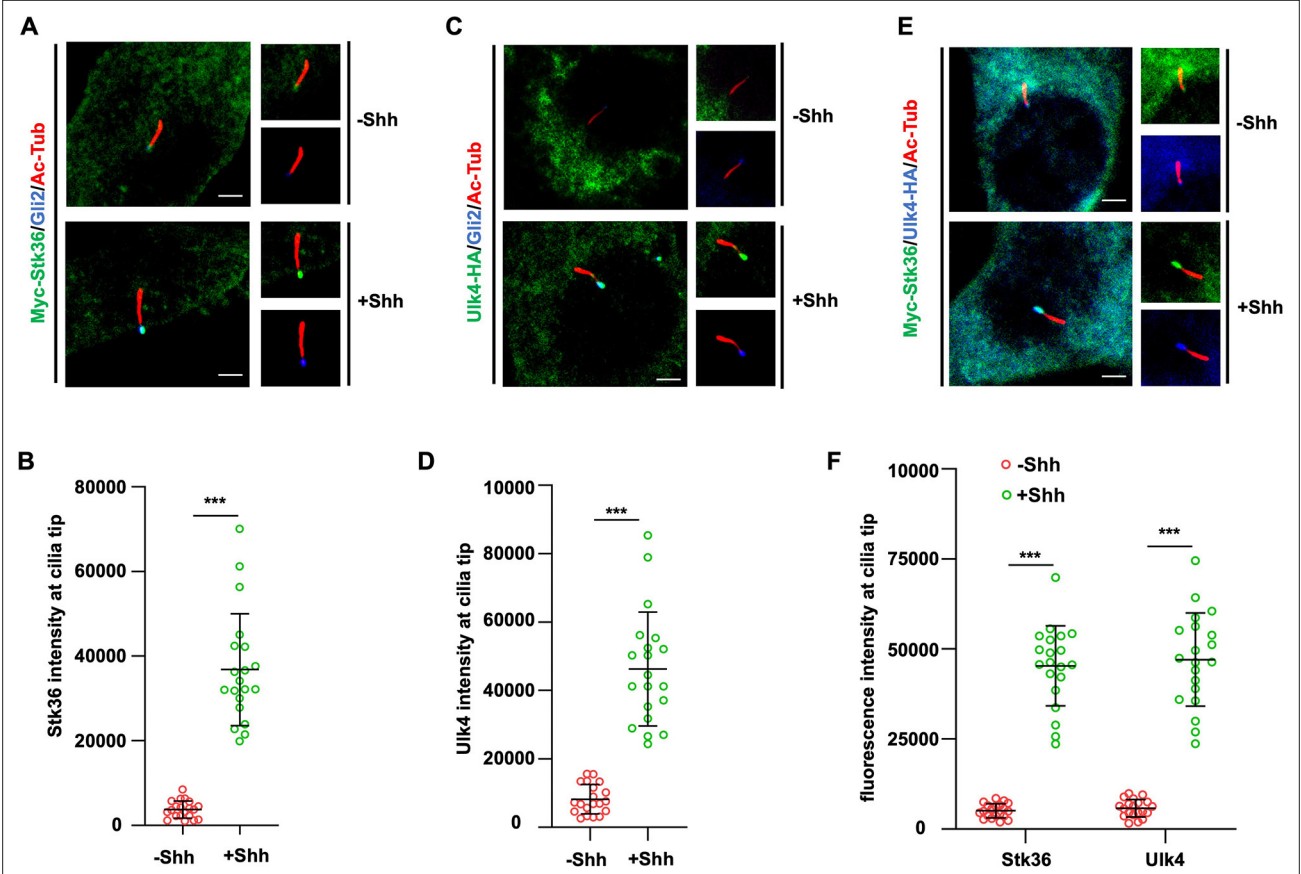

**Figure 5.** Shh stimulates ciliary tip accumulation of both Ulk4 and Stk36. (**A–D**) Representative images of immunostaining (**A, C**) and quantification (**B, D**) of ciliary tip localized Myc-Stk36 (green in **A**) or Ulk4-HA (green in **C**) and Gli2 (blue in **A** and **C**) in NIH3T3 cells infected with the Myc-Stk36 or Ulk4-HA lentivirus and treated with or without Shh-N. Primary cilia are marked by acetylated tubulin (Ac-tub) staining (red in **A** and **C**). (**E, F**) Representative images of immunostaining (**E**) and quantification (**F**) of ciliary tip localized Myc-Stk36 (green) and Ulk4-HA (blue) in NIH3T3 cells co-infected with Myc-Stk36 and Ulk4-HA lentiviruses and treated with or without Shh-N. Primary cilia are marked by Ac-Tub staining (red). The cells are starved in serum-free medium for 12 hr to allow ciliation and cultured in the same medium with or without Shh-N fragment for another 12 hr before they are subjected into immunostaining assay. The intensity of ciliary-localized Myc-Stk36, Ulk4-HA, and Gli2 was measured by ImageJ. Twenty cells were randomly selected and counted for each group. Data are mean ± SD from three independent experiments. ***p<0.001 (Student's *t*-test). Scale bars are 2 µM.

The online version of this article includes the following figure supplement(s) for figure 5:

**Figure supplement 1.** Shh stimulates ciliary tip accumulation of both Ulk4 and Stk36.

(*Figure 5—figure supplement 1G1–J2*). These observations suggest that Shh induces ciliary tip accumulation of Stk36-Ulk4-Gli2 protein complexes to promote Gli2 phosphorylation and activation.

## Ulk4 and Stk36 depend on each other for their ciliary localization

As pseudokinases could regulate the subcellular localization of their interacting kinases, we asked whether Ulk4 is required for ciliary localization of Stk36 in response to Shh. NIH3T3 cells stabling expressing Myc-Stk36 were depleted of Ulk4 by shRNA and treated with and without Shh. As shown in *Figure 6A and B*, Ulk4 RNAi diminished Shh-induced ciliary tip localization of Myc-Stk36, suggesting that Ulk4 is required for ciliary localization of Stk36 in response to Shh.

We next asked whether Ulk4 ciliary localization depends on Stk36. We first determined the ciliary location of Ulk4 in NIH3T3 cells with both Stk36 and Ulk3 knocked out (DKO) by the CRISPR-Cas9 system (*Han et al., 2019*). As shown in *Figure 6C and D*, Shh promoted ciliary tip enrichment of Ulk4-HA in the control cells but not in the Stk36 and Ulk3 DKO cells. Adding back wild type Stk36 (Stk36$^{WT}$) but not its kinase inactive form (Stk36$^{AA}$) restored Shh-induced Ulk4-HA ciliary tip accumulation in DKO cells (*Figure 6E and F*). By contrast, neither wild type Ulk3 (Ulk3$^{WT}$) nor its kinase dead

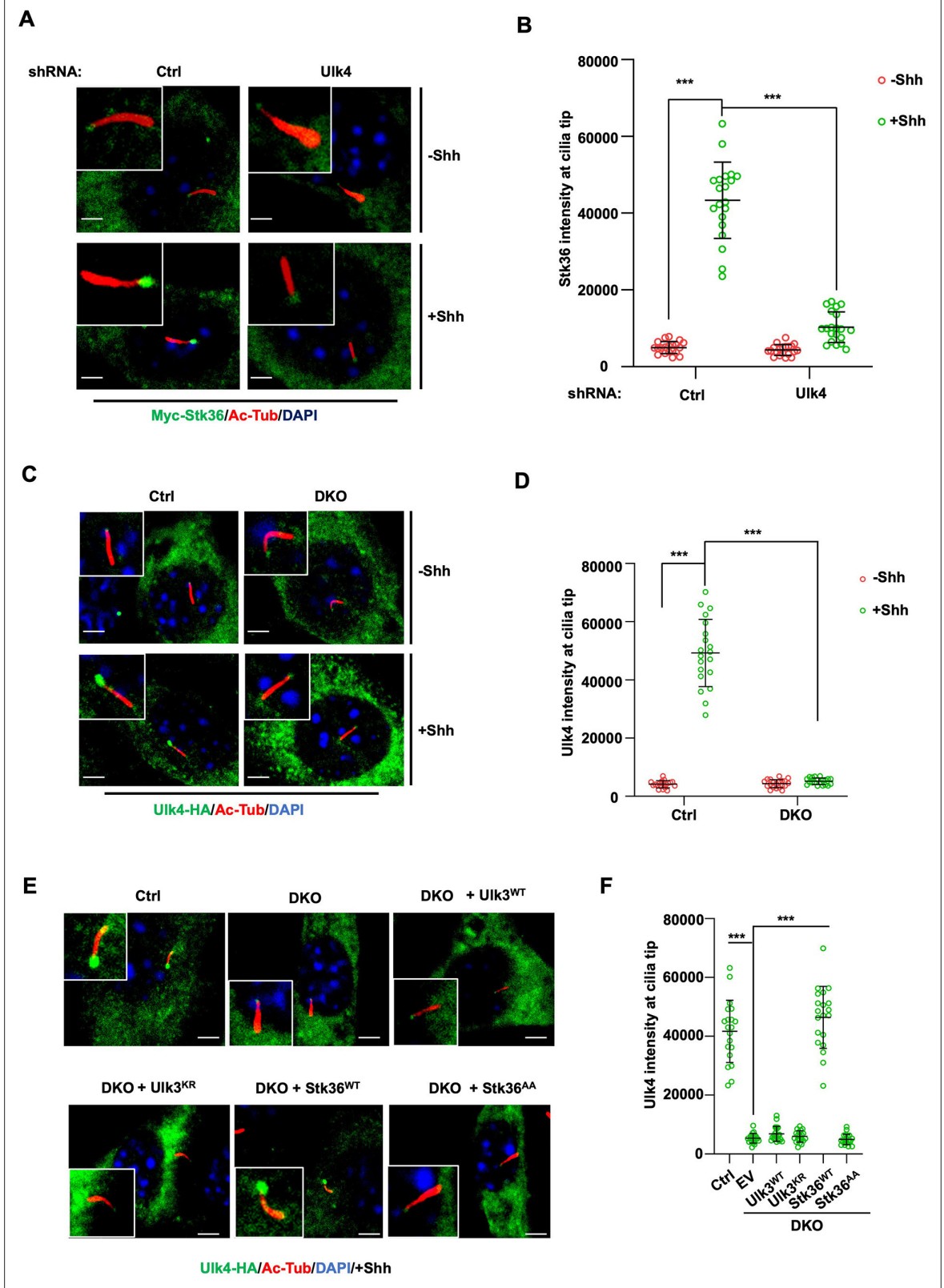

**Figure 6.** Stk36 and Ulk4 depend on each other for their ciliary tip accumulation. (**A, B**) Representative images of immunostaining (**A**) and quantification (**B**) of ciliary tip localized Myc-Stk36 in NIH3T3 cells infected with the indicated shRNA and Myc-Stk36 lentivirus in the presence or absence of Shh-N. Primary cilia are marked by Ac-Tub staining (red); Myc-Stk36 is marked by Myc staining (green); nuclei are marked by DAPI (Blue). (**C, D**) Representative images of immunostaining (**C**) and quantification (**D**) of ciliary tip localized Ulk4-HA (green) in wild type (Ctrl) or Ulk3 and Stk36 double knockout (DKO)

*Figure 6 continued on next page*

*Figure 6 continued*

NIH3T3 cells treated with or without Shh-N. Primary cilia are marked by Ac-Tub staining (red) and nuclei by DAPI (blue). (**E, F**) Representative images of immunostaining (**E**) and quantification (**F**) of ciliary-localized Ulk4-HA (green) in Shh-N-treated control or DKO NIH3T3 cells infected with or without lentiviruses expressing the indicated Ulk3 or Stk36 constructs. The cells are starved in serum-free medium for 12 hr to allow ciliation and cultured in the same medium with or without Shh-N fragment for another 12 hr before they are subjected into immunostaining assay. The intensity of ciliary-localized Myc-Stk36 or Ulk4-HA was measured by ImageJ. Twenty cells were randomly selected and counted for each group. Data are mean ± SD from three independent experiments. ***p<0.001 (Student's *t*-test). Scale bars are 2 μM.

form (Ulk3$^{KR}$) could rescue ciliary tip accumulation of Ulk4-HA in DKO cells (***Figure 6E and F***). These results suggest that ciliary tip localization of Ulk4 depends on Stk36 kinase activity.

## Ulk4 ciliary localization is regulated by its C-terminal phosphorylation

Many pseudokinases are also the substrates of their interacting kinases (***Kung and Jura, 2019***). To determine whether Ulk4 is a Stk36 substrate and whether Stk36-mediated phosphorylation of Ulk4 promotes its ciliary tip localization, we sought out to identify the relevant phosphorylation sites. We generated several C-terminally HA-tagged truncated forms of hUlk4 to narrow down the region essential for ciliary tip localization of Ulk4 (***Figure 7A***). These truncated forms were expressed in NIH3T3 cells in which endogenous Ulk4 depleted by RNAi. Deleting the entire regulatory domain abolished Shh-induced ciliary tip localization of the Ulk4 variant (hUlk4$_{1-450}$; ***Figure 7B and C***). Deleting the Stk36 binding region from Ulk4 (hUlk4$_{451-1275}$) also abolished its ciliary tip localization in response to Shh, suggesting that binding to Stk36 is essential for Ulk4 ciliary localization. We generated two small deletions from the C-terminus of hUlk4 (hUlk4$_{1-1000}$ and hUlk4$_{1-1200}$; ***Figure 7A***). Whereas hUlk4$_{1-1200}$ still localized to the ciliary tip in response to Shh, hUlk4$_{1-1000}$ failed to do so (***Figure 7B and C***), suggesting that the region between aa1000-1200 is essential for Shh-induced hUlk4 ciliary tip localization. Within this region, T1021 and T1088/T1090 fall into the Fu/Ulk3/Stk36 phosphorylation consensus site: S/T [X$_5$] E/D (***Figure 7D***; ***Han et al., 2019***; ***Zhou and Jiang, 2022***). Furthermore, phosphorylation on T1021 can prime CK1-mediated phosphorylation on T1023 as T1023 falls into the CK1 phosphorylation consensus site: D/E/$_{(p)}$S/$_{(P)}$T[X$_{1-3}$] S/T (***Knippschild et al., 2005***).

To determine whether phosphorylation on these sites may regulate Ulk4 ciliary localization, we generated hUlk4 variants carrying T1021A/1023A (hUlk4$^{T1021A/1023A}$) or T1088A/T1090A (hUlk4$^{T1088A/T1090A}$) substitutions. These hUlk4 variants were expressed in NIH3T3 cells with endogenous Ulk4 depleted by shRNA. We found that hUlk4$^{T1088A/T1090A}$ but not hUlk4$^{T1021A/1023A}$ accumulated at ciliary tip in response to Shh stimulation (***Figure 7E and F***), suggesting that phosphorylation on T1021/1023 is essential for Shh-induced ciliary tip localization of Ulk4.

To confirm T1021 phosphorylation by Stk36, we carried out an *in vitro* kinase assay using immuno-purified Fg-Stk36 as kinase and GST fusion protein containing hUlk4 sequence from aa1017 to aa1036 (GST-hUlk4C) as substrate. As shown in ***Figure 7G***, immunopurified wild type Stk36 (Fg-Stk36$^{WT}$) but not its kinase inactive variant (Fg-Stk36$^{AA}$) phosphorylated GST-hUlk4C. Mutating both T1021 and T1023 to A (T1021A /T1023A) abolished the phosphorylation; however, mutating T1021A reduced but not eliminated the phosphorylation, suggesting that in the absence of T1021 phosphorylation, T1023 could also be phosphorylated by Stk36. Interestingly, T1023 but not T1021 is conserved in mouse Ulk4 (mUlk4) and mUlk4 contains an Ala at T1021 (***Figure 7H***), implying that phosphorylation of T1021 might be dispensable for Ulk4 function. Indeed, we found that mutating T1021 to A in hUlk4 (hUlk4$^{T1021A}$) did not affect its ciliary tip accumulation in response to Shh (***Figure 7E and F***), as well as its activity in the Hh pathway (see below).

## Ciliary localization of Ulk4 is essential for its function in Shh signaling

To determine the importance of ciliary localization of Ulk4 for its function in Shh signal transduction, we compared the Hh pathway-related activity of ciliary localization-deficient hUlk4$^{T1021A/1023A}$ with that of ciliary localization-competent hUlk4$^{T1021A}$ (equivalent to mUlk4$^{WT}$) and hUlk4$^{WT}$ in NIH3T3 cells with endogenous Ulk4 depleted by shRNA. We found that both hUlk4$^{T1021A}$ and hUlk4$^{WT}$ rescued Shh-induced ciliary tip localization of coexpressed Myc-Stk36 (***Figure 8A and B***), phosphorylation of Myc-Gli2 at S230/S232 (***Figure 8C and C'***), and the expression of Shh target genes *Ptch1* and *Gli1* (***Figure 8D and E***). By contrast, hUlk4$^{T1021A/1023A}$ failed to rescue Shh-stimulated Stk36 ciliary tip localization, Gli2 phosphorylation, as well as *Ptch1* and *Gli1* expression in Ulk4-depleted cells (***Figure 8A–E***).

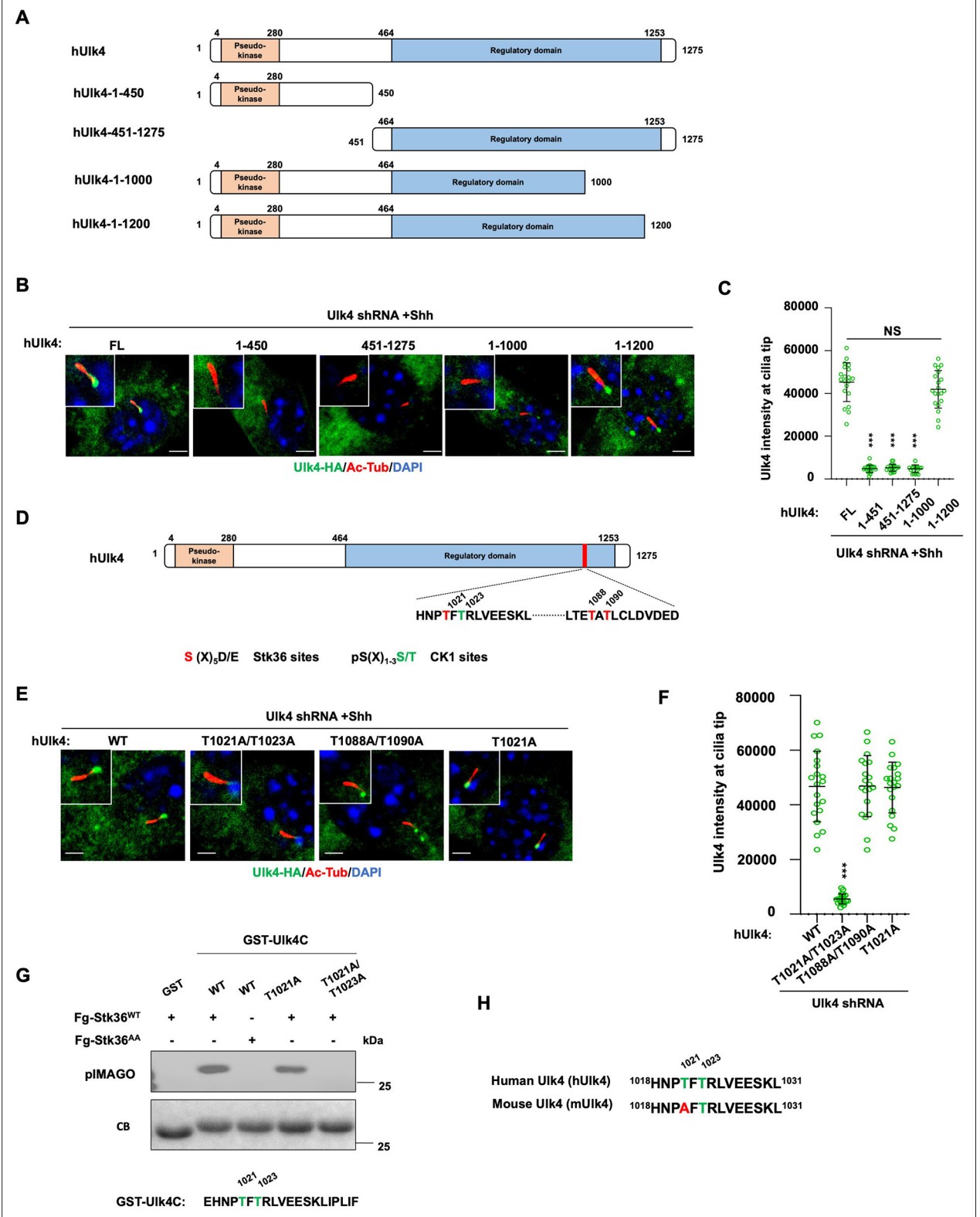

**Figure 7.** Ulk4 ciliary tip accumulation is promoted by its phosphorylation by Stk36. (**A**) Diagrams of hUlk4 deletion constructs. (**B, C**) Representative images of immunostaining (**B**) and quantification (**C**) of ciliary tip localized full-length or truncated hUlk4-HA in Ulk4 knockdown cells treated with Shh-N. (**D**) Stk36 and CK1 phosphorylation sites in the C-terminal region of hUlk4. (**E, F**) Representative images of immunostaining (**E**) and quantification (**F**) of ciliary tip localized wild type (WT) or mutant hUlk4-HA baring the indicated amino acid substitutions in Ulk4 knockdown NIH3T3 cells treated with

*Figure 7 continued on next page*

*Figure 7 continued*

Shh-N. (**G**) *In vitro* kinase assay using the immunopurified Fg-Stk36^WT or Fg-Stk36^AA as kinase and the indicated GST-Ulk4C fusion proteins as substrates. Phosphorylation was detected by the pIMAGO system. (**H**) Schematic diagram showing the sequence alignment of C-terminal phosphorylation sites of mouse Ulk4 (mUlk4) and human Ulk4 (hUlk4). The Thr residues phosphorylated by Stk36 are color-coded in green. The cells are starved in serum-free medium for 12 hr to allow ciliation and cultured in the same medium with or without Shh-N fragment for another 12 hr before they are subjected into immunostaining assay. The intensity of ciliary-localized WT and mutant Ulk4-HA was measured by ImageJ. Twenty cells were randomly selected and counted for each group. Data are mean ± SD from three independent experiments. ***p<0.001 (Student's *t*-test). Scale bars are 2 µM.

The online version of this article includes the following source data for figure 7:

**Source data 1.** Source data for western blots in *Figure 7G*.

These results suggest that ciliary localization of Ulk4 is essential for its ability to promote Stk36 ciliary localization, Gli2 phosphorylation, and Hh pathway activation (*Figure 8F*).

## Discussion

The Gli family of transcription factors mediates the transcriptional output of the Hh signaling pathway in species ranging from *Drosophila* to mammals but how Gli is activated by Hh signaling is still poorly understood, especially in vertebrates where Gli activation is thought to occur at the tip of primary cilium. We recently found that the Fu/Ulk family kinases directly phosphorylate Ci and Gli2 on multiple sites to promote their activation (*Han et al., 2019*; *Zhou et al., 2022*; *Zhou and Jiang, 2022*). In mammalian cells, the Ulk family kinases Ulk3 and Stk36 (mammalian homolog of Fu) act in parallel to promote Gli2 phosphorylation and activation in response to Shh (*Han et al., 2019*). In addition, Gli2 phosphorylation depends on its ciliary localization (*Han et al., 2019*). In this study, we found that Stk36 accumulated to the tip of primary cilium and colocalized with Gli2 in response to Shh stimulation, consistent with Gli2 being converted into Gli2^A at the ciliary tip. In addition, we identified another Ulk family member Ulk4, which is a pseudokinase, as a mediator of Shh-induced Gli2 phosphorylation. We found that Ulk4 and Stk36 act in the same pathway but in parallel with Ulk3 to promote Gli2 phosphorylation and activation in NIH3T3 and MEF cells. The positive role of Ulk4 in Shh signaling could explain why elevated expression of Ulk4 in certain genetic background could suppress the heart outflow tract defects and holoprosencephaly caused by partially reduced Shh signaling activity in LRP2-deficient mice (*Mecklenburg et al., 2021*). However, it remains to be determined whether Ulk4 participates in Hh signaling *in vivo* by loss-of-function study. Because knockout of Stk36 in mice did not affect Hh signaling during development, likely due to functional compensation by other kinases such as Ulk3 (*Chen et al., 2005*; *Merchant et al., 2005*), we do not expect that loss of function of Ulk4 will affect Hh signaling during development. Therefore, double knockout of Ulk4 (or Stk36) and Ulk3 will be necessary to address the physiological role of the Fu/Ulk family members in mammalian Hh signaling. In addition, it will also be important to determine whether Ulk4/Stk36 and Ulk3 could be differentially required for Hh signal transduction in different cell types/tissues or in different contexts, for example, development vs tissue homeostasis or physiological vs pathological settings.

To determine the mechanism by which Ulk4 assists Stk36 to phosphorylate Gli2, we explored whether Ulk4 regulated (1) the enzymatic activity of Stk36; (2) Shh-induced Stk36 kinase activation; (3) the association of Stk36 and its substrate; and (4) ciliary localization Stk36. In an *in vitro* kinase assay, we found that immunopurified Stk36 phosphorylated GST-Gli2N equally well regardless of whether it interacted with Ulk4 or not, suggesting that Ulk4 does not regulate Stk36 kinase activity. Rather, we found that Ulk4 is required for Stk36 to bind to its substrate Gli2. Ulk4 interacts with Stk36 through its N-terminal region that contains the pseudokinase domain and with Gli2 via its C-terminal regulatory domain. Depletion of Ulk4 diminished the physical association between Stk36 and Gli2. As a further support of the notion that Gli2 is brought to Stk36 through Ulk4, we found that the kinase domain of Stk36, which binds Ulk4, also interacted with Gli2, whereas its C-terminal region interacted with neither Ulk4 nor Gli2. Of note, despite being a pseudokinase, Ulk4 contains an unusual ATP binding site that might regulate its ability to bind other proteins such as Stk36 (*Preuss et al., 2020*). Therefore, it would be interesting to determine in the future whether ATP binding to Ulk4 could regulate Shh signaling or other processes that Ulk4 is involved.

Although Stk36/Ulk4/Gli2 complex formation appears to be constitutive regardless of the presence or absence of Shh, Stk36 kinase activation depends on Shh stimulation. Like Fu, Stk36 is

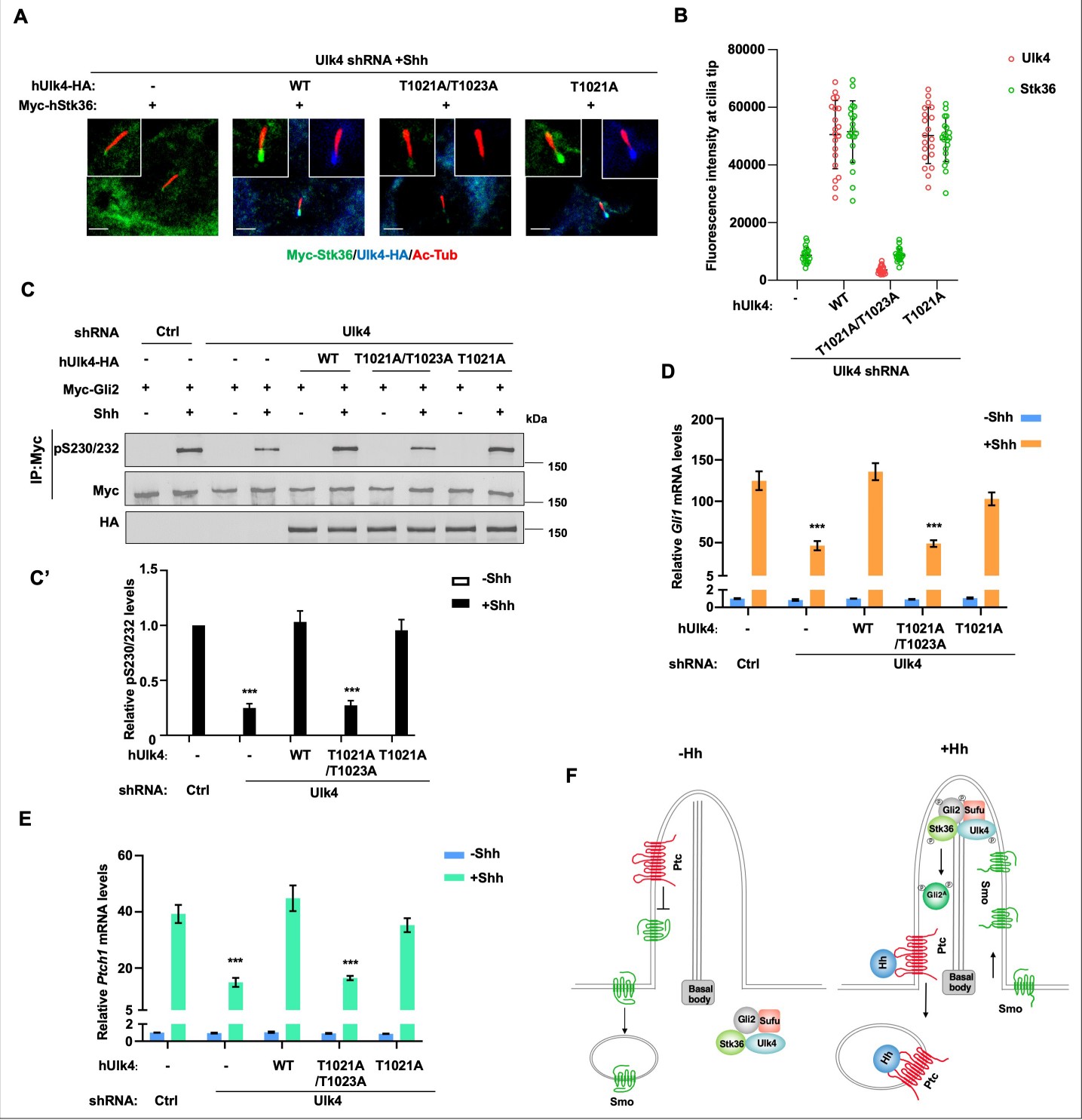

**Figure 8.** Ciliary tip localization of Ulk4 is required for Shh signal transduction. (**A, B**) Representative images of immunostaining (**A**) and quantification (**B**) of ciliary tip localized Myc-Stk36 and hUlk4 in Ulk4-depleted NIH3T3 cells expressing the indicated hUlk4 constructs and treated with Shh. (**C, C'**) Western blot analysis (**C**) and quantification (**C'**) of Myc-Gli2 phosphorylation on S230/S232 in NIH3T3 cells expressing the indicated shRNA and hUlk4 lentiviral constructs and treated with or without Shh-N. Data are mean ± SD from two independent experiments. ***p<0.001 (Student's *t*-test). (**D, E**) Relative *Gli1* (**D**) and *Ptch1* (**E**) mRNA levels in NIH3T3 cells expressing the indicated shRNA and hUlk4 lentiviral constructs treated with or without Shh-N. (**F**) Model for how Ulk4 participates in Shh signaling. See text for details. The cells are starved in serum-free medium for 12 hr to allow ciliation and cultured in the same medium with or without Shh-N fragment for another 12 hr before they are subjected into RNA preparation, western blot, and

*Figure 8 continued on next page*

*Figure 8 continued*

immunostaining assay. The intensity of ciliary-localized WT and mutant Ulk4-HA was measured by ImageJ. Twenty cells were randomly selected and counted for each group. Data are mean ± SD from three independent experiments. ***p<0.001 (Student's *t*-test). Scale bars are 2 µM.

The online version of this article includes the following source data for figure 8:

**Source data 1.** Source data for western blots in *Figure 8C*.

phosphorylated on multiple sites in the activation loop of its kinase domain in response to Shh likely through trans-autophosphorylation, and activation loop phosphorylation is essential for its kinase activation. Using a phospho-specific antibody that recognized pT158/pS159 in the kinase activation loop of Fu/Stk36, we found that Ulk4 depletion did not affect Shh-induced phosphorylation of Stk36 on T158/S159, a readout for Stk36 kinase activation, suggesting that Ulk4 is dispensable for Shh signaling to activate Stk36. In *Drosophila*, the kinesin-like protein Costal2 (Cos2) mediates the activation of Fu kinase by Hh. Therefore, it would be interesting to determine whether mammalian homologs Cos2, Kif7, and/or Kif27 participate in Stk36 kinase activation in response to Shh stimulation.

Hh signaling depends on primary cilia in vertebrates. Upon Shh stimulation, key pathway components such as Smo and Gli proteins accumulate in primary cilia, and Gli2 is accumulated to the ciliary tip and thought to be converted into its activator form (Gli2$^A$) in this ciliary compartment. Consistent with this notion, we found that Shh induced ciliary accumulation of both Ulk4 and Stk36 that colocalized with Gli2 at the ciliary tip. Furthermore, we found that Ulk4 is required for ciliary tip accumulation of Stk36. On the other hand, ciliary tip accumulation of Ulk4 depends on Stk36 kinase activity and phosphorylation of its C-terminal region on T1023. Preventing Ulk4 phosphorylation on T1023 abolished its ciliary tip accumulation and its ability to promote Stk36 ciliary tip accumulation, Gli2 phosphorylation, and Hh pathway activation. These results suggest that ciliary tip accumulation of Ulk4 is critical for its function in Hh signal transduction, likely by increasing the local concentration of Stk36 and Gli2 to promote effective phosphorylation of Gli2. It would also be important to examine the localization of endogenous Ulk4 and Stk36 in the future as exogenously expressed epitope-tagged protein might cause artifacts. In addition, the precise mechanism and dynamics of Ulk4 ciliary targeting remain to be determined in the future. For example, it would be important to determine how phosphorylation of T1023 promotes ciliary accumulation of Ulk4. A recent study identified several motor proteins of the kinesin family (KIF1B, KIF3A, KIF3B, and KAP3) as the interaction partners of Ulk4 (*Preuss et al., 2020*). KIF3A, KIF3B, and KAP3 are components of kinesin2 motor that act in conjunction with intraflagellar transport (IFT) complexes to transport cargoes into primary cilia (*Goetz and Anderson, 2010*). It would be interesting to determine whether Ulk4 T1023 phosphorylation affects its interaction with KIF3A, KIF3B, KAP3, or components in the IFT complexes. Of note, a previous study suggested that Ulk4 regulated the ciliary localization of Smo, although the underlying mechanism was not explored (*Mecklenburg et al., 2021*). Therefore, it remains possible that Ulk4 could regulate mammalian Hh signaling at multiple steps similar to what Fu does in the *Drosophila* Hh pathway (*Claret et al., 2007*; *Liu et al., 2007*).

Both Ulk4 and Stk36 have functions outside the Hh pathway. Stk36 is required for motile ciliogenesis, and its loss of function affects the construction of central pair apparatus of '9+2' motile cilia in both zebrafish and mice (*Wilson et al., 2009*; *Nozawa et al., 2013*). Ulk4 malfunction also affects motile ciliogenesis of ependymal cells in mutant mice, leading to hydrocephalus (*Vogel et al., 2012*; *Liu et al., 2016b*). A recent study demonstrated that Ulk4 and Stk36 physically interact to mediate the assembly of motile flagella in the flagellated protist *Leishmania mexicana* (*McCoy et al., 2023*). However, the biochemical mechanism by which Ulk4/Stk36 regulates motile ciliogenesis has remained unknown. Ulk4 could regulate Stk36 ciliary localization and/or kinase–substrate interaction like what we have discovered for their role in Shh signaling. The relevant substrates for Stk36 in motile ciliogenesis remain to be determined. Knowing the phosphorylation consensus sites for the Fu/Ulk family kinase may facilitate the identification of Stk36 substrates involved in the assembly of motile cilia and flagella. Initially identified Fu/Ulk3/Stk36 phosphorylation sites in Ci/Gli and Cos2 fall into the consensus: S/T(X)$_5$D/E, where the acidic residue at +6 position is essential (*Han et al., 2019*; *Zhou and Jiang, 2022*). However, subsequent studies suggested that the spacing between the phospho-acceptor site and the acid residue appears to be flexible. For example, a Ulk3/Stk36 phosphorylation site in the C-terminal region of Gli2 has a spacing of six residues between the phospho-acceptor site and the acid residue (*Zhou et al., 2022*). Here, we found that an Stk36 phosphorylation site in the

C-terminal region of Ulk4 contains only four residues between the phospho-acceptor site and the acid residue. Based on these findings, we propose an expanded consensus site for the Fu/Ulk family kinases as $S/T(X)_{4-6}D/E$. It would be interesting to determine whether this phosphorylation consensus sequence could be extended to other members of the Ulk kinase family such as Ulk1/Atg1 and Ulk2, which are the initiating kinases for autophagy (*Zachari and Ganley, 2017*).

Ulk4 has been implicated in several human diseases, including hypertension and psychiatric disorder such as schizophrenia (*Luo et al., 2022*). Ulk4 also plays a role in neurogenesis in mice by regulating neural stem cell pool and neurite branching morphogenesis (*Lang et al., 2014*; *Lang et al., 2016*; *Liu et al., 2016a*). In addition, Ulk4 regulates the integrity of white matter by promoting myelination (*Liu et al., 2018b*). How Ulk4 participates in these diverse processes has remained unclear. A recent study revealed that Ulk4 binds several other kinases and phosphatases such as DAPK3, LMTK2, PTPN14, as well as proteins implicated in the regulation of neuronal differentiation and axonal regeneration, including CAMSAP family proteins CAMSAP1/3 (*Pongrakhananon et al., 2018*; *Preuss et al., 2020*). Given our findings regarding how Ulk4 regulates Hh signaling, it is tempting to speculate that Ulk4 may also function as a scaffold to bridge kinases or other enzymes to their substrates and/or regulate the subcellular localization of the interacting proteins in other biological processes.

## Materials and methods

### DNA constructs

Myc-Gli2 and GST-Gli2 were described previously (*Han et al., 2019*). All GST-fusion (the GST-tagged human Ulk4C[WT], Ulk4C[T1021A], Ulk4C[T1021A/1023A]) proteins were subcloned into the *pGEX 4T-1* vector using EcoRI and XhoI sites. N-terminally CFP-tagged mouse Ulk3[WT], Ulk3[KR], N-terminally 6XMyc-tagged human Stk36, N-terminally 3XFg-tagged human Stk36[WT], Stk36[AA] and the C-terminally 3XHA-tagged human Ulk4[WT], Ulk4[1-451], Ulk4[452-1275], Ulk4[1-1000], Ulk4[1-1200], Ulk4[T1021/1023A], Ulk4[T1088/1090A], Ulk4[T1021A] were subcloned into the *FUGW* vector with the EcoRI and BamHI sites. N-terminally 3XFg-tagged human Stk36[WT], Stk36[EE], Stk36[AA], Stk36[1-260], Stk36[261-1315], and C-terminally 3XHA-tagged human Ulk4[WT], Ulk4[1-451], Ulk4[452-1275] were subcloned in the *PLVX* vectors using the EcoRI and MluI sites. The Ulk4[T1021/1023A], Ulk4[T1088/1090A], Ulk4[T1021A], Stk36[EE], Stk36[AA] were generated by PCR-mediated site-directed mutagenesis.

### Cell culture, transfection, and lentiviral production

NIH3T3 cells (ATCC, CRL-1658) were cultured in Dulbecco's modified Eagle's medium (DMEM, Sigma) containing 10% bovine calf serum (Gibco) and 1% penicillin/streptomycin (Sigma). The transfection was carried out by using the GenJet Plus *in vitro* DNA transfection kit according to the manufacturer's instruction (SignaGen). HEK293T cells (ATCC, CRL-11268) were cultured in DMEM (Sigma) supplemented with 10% fetal bovine serum (Gibco) and 1% penicillin/streptomycin (Sigma). PolyJet *in vitro* DNA transfection kit (SignaGen) was used to do the transfection according to the manufacturer's instruction. MEF cells (ATCC, CRL-2991) were cultured in DMEM (Sigma) containing 10% FBS (Gibco) and 1% streptomycin/penicillin (Sigma) according to the standard protocol. For Shh-N treatment, cells were cultured at 50%-confluent densities in plate or chamber slides for 1 d and starved in serum-free medium (0.5% BCS for NIH3T3 cells or 0.5% FBS for MEF cells) for 12 hr to allow ciliation. The recombinant human Shh N-terminal fragment (1 ng/ml; R&D Systems, #8908-SH-005) was then added to the same serum-free medium for 12 hr or overnight.

### Immunostaining, immunoprecipitation, and western blot analysis

The protocol for Ulk4-HA/Myc-Stk36 immunostaining was carried out as previously described (*Han et al., 2019*). Immunofluorescence imaging was captured using a Zeiss LSM700 laser scanning microscope. For ciliary Myc-Stk36/Ulk4-HA quantification, the cilia were labeled with the acetylated-Tubulin and the Myc-Stk36/Ulk4-HA fluorescence intensities were measured in the Myc-Stk36/Ulk4-HA image channel by ImageJ. Twenty cells were randomly selected and calculated for each group. For immunoprecipitation assay, after transfection for 48 hr, cells were washed twice with PBS and then lysed on ice for 10 min with lysis buffer containing 1 M Tris pH 8.0, 5 M NaCl, 1 MNaF, 0.1 M $Na_3VO_4$, 1% CA630, 10% glycerol, and 0.5 M EDTA (pH 8.0). Cell lysates were incubated with protein A-Sepharose beads (Thermo Scientific) for 1 hr at 4°C to eliminate nonspecific binding proteins. After removal of the protein-A beads by centrifugation, the cleared lysates were incubated with Myc (HA or Flag)

antibody for 4 hr or overnight. Protein complexes were collected by incubation with protein A-Sepharose beads for 1 hr at 4°C, followed by centrifugation. Immunoprecipitates were washed three times for 5 min each with lysis buffer and were separated on SDS-PAGE. Western blot was carried out using standard protocol. Western blot quantification was carried out by employing fluorogenic secondary antibody (IRDye 680LT goat anti-mouse: 926-68020 or goat anti-rabbit: 926-68021), and the blot was scanned and quantified with an Odyssey CLx imaging system (LI-COR). The antibodies used in this study were as follows: mouse anti-Myc (Santa Cruz Biotechnology; 9E10), mouse anti-Flag (Sigma; M2), mouse anti-HA (Santa Cruz Biotechnology; F7), mouse anti-acetylated Tubulin (Sigma; T7451; staining:1:1000), rabbit anti-Myc (Abcam; ab9106; staining 1:2000), rabbit anti-HA (Abcam; ab9110; staining 1:500), and goat anti-Gli2 (R&D Systems; AF3635; staining 1:1000).

## *In vitro* kinase assay

*In vitro* kinase assay was performed by incubating 25 µl of reaction mixtures containing 150 mM Tris-HCl (pH 7.5), 0.2 mM $Mg^{2+}$/ATP, 1 µg of purified GST-Ulk4/Gli2 fusion proteins, together with appropriate amount of kinase for 30 min at 30°C. The reaction was terminated by adding 2× SDS loading buffer. The resultant samples were loaded on SDS-PAGE and subjected to pIMAGO phosphoprotein detection kit with fluor-680 (Sigma) detection kit and LI-COR Odyssey platform for western blot analyses of phosphorylated proteins (MilliporeSigma). Fg-Stk36$^{WT}$ and Fg-Stk36$^{AA}$ were immunopurified from HEK293 cells transfected with the Fg-Stk36$^{WT}$ or Fg-Stk36$^{AA}$ construct.

## Viral infection and shRNA

For lentivirus generation, *pLKO.1* vectors expressing shRNA or *FUGW* vectors expressing transgenes were co-transfected with package vectors (PSPAX2 and PMDG2) into HEK293T cells using PolyJet (SignaGen). The supernatants were collected after 48 hr and 72 hr transfection, filtered through a 0.45 mM filter, and centrifuged at 20,000 × *g* for 2 hr. The sediment containing the viruses was resuspended, aliquoted in small volume of culture medium, and stored at –80°C for future use. For cell infection, lentiviruses were added to 70% confluent wild type NIH3T3 or MEF cells together with Polybrene (Sigma, Cat# H9268) overnight to make the stable NIH3T3 or MEF cell lines expressing transgene or shRNA. The mouse Ulk4 shRNA lentiviral vector was purchased from Sigma (TRCN0000328268 for Ulk4 knockdown in NIH3T3 cells and MEFs; TRCN0000002203 for Ulk4 knockdown in HEK293 cells). The non-target shRNA plasmid DNA (Sigma; SHC016) was used as control.

## RNA interference

For RNAi experiments in NIH3T3 or MEF cells, the siRNAs were purchased from Sigma. The sequences for Ulk3 silencing were 5'-GUGUACAAGGCCUACGCCA-3' (Cat# SASI_ Mm02 _ 00444409) and 5'-CUGAGAAGGUGGCCCGUGU-3' (Cat# SASI_Mm02_00444410). The sequences for Stk36 silencing were 5'- GGUAUACUGGCUUCAGAAA-3' (Cat# SASI_ Mm02 _00345637) and 5'-GCCUUAUGUGCU UUGCUGU-3' (Cat# SASI_Mm01_00167751). The target sequence of mouse Ulk4 shRNA is CTGC GAAGATTATCGAGAATG. The target sequence of human Ulk4 shRNA is CCACTAGGTCACTCTT TCAGA. The sequences for negative control were 5'-UUCUCCGAACGUGUCACG U-3'. The RNAi MAX reagent (Invitrogen, Cat# 13778150) was used for the transfection of siRNA. The knockdown efficiency was validated via real-time PCR.

## Quantitative RT-PCR

Total RNA was extracted from $1 \times 10^6$ cells using the RNeasy Plus Mini Kit (QIAGEN), and cDNA was synthesized with High-Capacity cDNA Reverse Transcription Kit (Applied Biosystems; 01279158) and qPCR was performed using Fast SYBR Green Master Mix (Applied Biosystems; 2608088) and a Bio-Rad CFX96 real-time PCR system (Bio-Rad; HSP9601). The RT-qPCR was performed in triplicate for each of three independent biological replicates. Quantification of mRNA levels was calculated using the comparative CT method. Primers: Gli1, 5'-GTGCACGTTTGAAGGCTGTC-3' and 5'-GAGTGGGT CCGATTCTGGTG-3'; Ptch1, 5'-GAAGCCACAGAAAACCCTGTC-3' and 5'-GCCGCAAGCCTTCTCT ACG-3'; Ulk3, 5'- ACGAAACATCTCTCACTTG-3' and 5'-TGCTGGGCAAAGCCAAAGTC-3'; Ulk4, 5'-ATGCAGAGTGTGATTGCGTTG-3' and 5'-GAGTGGCAGTTTCTGTGAACA-3'; Stk36, 5'- CGCATCCT ACACCGAGATATGA-3' and 5'-AAATCCAAAGTCACAGAGCTTGA-3'; GAPDH, 5'-GTGGTGAAGCAG GCATCTGA-3' and 5'-GCCATGTAGGCCATGAGGTC-3'.

## Acknowledgements

We thank Bing Wang, Xu Li, and Yong Cho for technical assistance, Pancheng Xie and Yi Liu for reagents. J Jiang is a Eugene McDermott Endowed Scholar in Biomedical Science at the University of Texas Southwestern.

## Additional information

### Funding

| Funder | Grant reference number | Author |
|---|---|---|
| National Institute of General Medical Sciences | R35GM118063 | Jin Jiang |
| Welch Foundation | I-1603 | Jin Jiang |

The funders had no role in study design, data collection and interpretation, or the decision to submit the work for publication.

### Author contributions

Mengmeng Zhou, Conceptualization, Data curation, Software, Formal analysis, Validation, Investigation, Visualization, Methodology, Writing - original draft, Writing - review and editing; Yuhong Han, Conceptualization, Investigation; Jin Jiang, Conceptualization, Resources, Funding acquisition, Writing - original draft, Writing - review and editing

### Author ORCIDs

Mengmeng Zhou ⓘ http://orcid.org/0000-0002-4945-8632
Jin Jiang ⓘ http://orcid.org/0000-0002-5951-372X

Reviewer #1 (Public Review): https://doi.org/10.7554/eLife.88637.3.sa1
Reviewer #2 (Public Review): https://doi.org/10.7554/eLife.88637.3.sa2
Reviewer #3 (Public Review): https://doi.org/10.7554/eLife.88637.3.sa3
Author Response https://doi.org/10.7554/eLife.88637.3.sa4

## Additional files

### Supplementary files

• MDAR checklist

### Data availability

Source data files have been provided for Figure 1, Figure 2, Figure 3, Figure 4, Figure 7 and Figure 8.

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
