## [Editor Report · eLife assessment]

This **fundamental** study substantially advances our understanding of how the pseudokinase ULK4 interacts with an active member of the same kinase subfamily (STK36) to promote GLI phosphorylation and Hedgehog pathway activation. The evidence supporting the proposed mechanism is **compelling**, with rigorous biochemical assays and state-of-the-art cell-based imaging techniques. The work will be of broad interest to cell biologists and biochemists.

---

## [Referee Report · Reviewer #1 (Public Review)]

In their study, Zhou et al. unveil the pivotal role of ULK4 in conjunction with STK36, shedding light on their collective impact on GLI2 phosphorylation and the subsequent activation of the SHH pathway. The research delves deep into the intricate interactions between ULK4 and various components of the SHH pathway within the primary cilium.

The main strength of the study lies in the careful and systematic sequence of logical methods. The authors apply the expression of a range of different deletion and mutation constructs and carry out a comprehensive biochemical study of the consequences of depletion and reintroduction of various components in the context of STK36 and ULK4.

Their findings reveal that ULK4 forms dynamic interactions with a complex composed of STK36 and GLI2. It is proposed that ULK4 acts as a scaffold, facilitating the essential interaction between STK36 and GLI2, thereby driving GLI2 phosphorylation by STK36. Notably, the research reveals that the N-terminal pseudokinase domain of ULK4 binds to Stk36, while the C-terminal regulatory domain of ULK4 interacts with Gli2. Moreover, the study presents compelling evidence for co-localization of ULK4 and STK36 with GLI2 at the ciliary tip within NIH 3T3 cells. Importantly, ULK4 and STK36 mutually rely on each other for their accumulation at this ciliary tip.

This intricate mechanism, orchestrated by ULK4, brings to light the nuanced modulation of the SHH pathway. The research is substantiated by rigorous Co-IP experiments, kinase assays, and confocal imaging localization studies. To unravel the fine details of GLI2 phosphorylation at the primary cilium tip, the authors meticulously employ a diverse array of mutated and wild-type constructs of STK36 and ULK4.

In summary, the studiy provide compelling insights into the intricate regulation of signaling pathways. Zhou et al.'s work on ULK4 and STK36 in the SHH pathway deepen our understanding of these complex processes, offering potential avenues for drug development, particularly in the context of cancer therapeutics.

---

## [Referee Report · Reviewer #2 (Public Review)]

The authors provide solid molecular and cellular evidence that ULK4 and STK36 not only interact, but that STK36 is targeted (transported?) to the cilium by ULK4. Their data helps generate a model for ULK4 acting as a scaffold for both STK36 and its substrate, Gli2, which appear to co-localise through mutual binding to ULK4. This makes sense, given the proposed role of most pseuodkinases as non-catalytic signaling hubs. There is also an important mechanistic analysis performed, in which ULK4 phosphorylation in an acidic consensus by STK36 is demonstrated using IP'd STK36 or an inactive 'AA' mutant, which suggests this phosphorylation is direct.

The major strength of the study is the well-executed combination of logical approaches taken, including expression of various deletion and mutation constructs and the careful (but not always quantified in immunoblot) effects of depleting and adding back various components in the context of both STK36 and ULK3, which broadens the potential impact of the work. The biochemical analysis of ULK4 phosphorylation appears to be solid, and the mutational study at a particular pair of phosphorylation sites upstream of an acidic residue (notably T2023) is further strong evidence of a functional interaction between ULK4/STK36. The possibility that ULK4 requires ATP binding for these mechanisms is not approached, though would provide significant insight: for example it would be useful to ask if Lys39 in ULK4 is involved in any of these processes, because this residue is likely important for shaping the ULK4 substrate-binding site as a consequence of ATP binding; this was originally shown in PMID 24107129 and discussed more recently in PMID: 33147475 in the context of the large amount of ULK4 proteomics data released.

The discussion is excellent, and raises numerous important future work in terms of potential transportation mechanisms of this complex. It also explains why the ULK4 pseudokinase domain is linked to an extended C-terminal region. Does AF2 predict any structural motifs in this region that might support binding to Gli2?

A weakness in the study, which is most evident in Figure 1, where Ulk4 siRNA is performed in the NIH3T3 model (and effects on Shh targets and Gli2 phosphorylation assessed), is that we do not know if ULK4 protein is originally present in these cells in order to actually be depleted. Also, we are not informed if the ULK4 siRNA has an effect on the 'rescue' by HA-ULK4; perhaps the HA-ULK4 plasmid is RNAi resistant, or if not, this explains why phosphorylation of Gli2 never reaches zero? Given the important findings of this study, it would be useful for the authors to comment on this, and perhaps discuss if they have tried to evaluate endogenous levels of ULK4 (and Stk36) in these cells using antibody-based approaches, ideally in the presence and absence of Shh. The authors note early on the large number of binding partners identified for ULK4, and siRNA may unwittingly deplete some other proteins that could also be involved in ULK4 transport/stability in their cellular model.

The sequence of ULK4 siRNAs is not included in the materials and methods as far as I can see, though this is corrected in the next version of the manuscript.

---

## [Referee Report · Reviewer #3 (Public Review)]

In this manuscript, Zhou et al. demonstrate that the pseudokinase ULK4 has an important role in Hedgehog signaling by scaffolding the active kinase Stk36 and the transcription factor Gli2, enabling Gli2 to be phosphorylated and activated.

Through nice biochemistry experiments, they show convincingly that the N-terminal pseudokinase domain of ULK4 binds Stk36 and the C-terminal Heat repeats bind Gli2.

Lastly, they show that upon Sonic Hedgehog signaling, ULK4 localizes to the cilia and is needed to localize Stk36 and Gli2 for proper activation.

This manuscript is very solid and methodically shows the role of ULK4 and STK36 throughout the whole paper, with well controlled experiments. The phosphomimetic and incapable mutations are very convincing as well.

I think this manuscript is strong and stands as is, and there is no need for additional experiments.

Overall, the strengths are the rigor of the methods, and the convincing case they bring for the formation of the ULK4-Gli2-Stk36 complex. There are no weaknesses noted. I think a little additional context for what is being observed in the immunofluorescence might benefit readers who are not familiar with these cell types and structures.

The revised manuscript has improved some of the unclear areas.

---

## [Author Response]

The following is the authors’ response to the original reviews.

**Reviewer #1 (Public Review):**
The Hedgehog (HH) protein family is important for embryonic development and adult tissue maintenance. Deregulation or even temporal imbalances in the activity of one of the main players in the HH field, sonic hedgehog (SHH), can lead to a variety of human diseases, ranging from congenital brain disorders to diverse forms of cancers. SHH activates the GLI family of transcription factors, yet the mechanisms underlying GLI activation remain poorly understood. Modification and activation of one of the main SHH signalling mediators, GLI2, depends on its localization to the tip of the primary cilium. In a previous study the lab had provided evidence that SHH activates GLI2 by stimulating its phosphorylation on conserved sites through Unc-51-like kinase 3 (ULK3) and another ULK family member, STK36 (Han et al., 2019). Recently, another ULK family member, ULK4, was identified as a modulator of the SHH pathway (Mecklenburg et al. 2021). However, the underlying mechanisms by which ULK4 enhances SHH signalling remained unknown. To address this question, the authors employed complex biochemistry-based approaches and localization studies in cell culture to examine the mode of ULK4 activity in the primary cilium in response to SHH. The study by Zhou et al. demonstrates that ULK4, in conjunction with STK36, promotes GLI2 phosphorylation and thereby SHH pathway activation. Further experiments were conducted to investigate how ULK4 interacts with SHH pathway components in the primary cilium. The authors show that ULK4 interacts with a complex formed between STK36 and GLI2 and hypothesize that ULK4 functions as a scaffold to facilitate STK36 and GLI2 interaction and thereby GLI2 phosphorylation by STK36. Furthermore, the authors provide evidence that ULK4 and STK36 co-localize with GLI2 at the ciliary tip of NIH 3T3 cells, and that ULK4 and STK36 depend on each other for their ciliary tip accumulation. Overall, the described ULK4-mediated mechanism of SHH pathway modulation is based on detailed and rigorous Co-IP experiments and kinase assays as well as confocal imaging localization studies. The authors used various mutated and wild-type constructs of STK36 and ULK4 to decipher the mechanisms underlying GLI2 phosphorylation at the tip of the primary cilium. These novel results on SHH pathway activation add valuable insight into the complexity of SHH pathway regulation. The data also provide possible new strategies for interfering with SHH signalling which has implications in drug development (e.g., cancer drugs).However, it will be necessary to explore additional model systems, besides NIH3T3, HEK293 and MEF cell cultures, to conclude on the universality of the mechanisms described in this study. Ultimately, it needs to be addressed whether ULK4 modulates SHH pathway activity *in vivo*. Is there evidence that genetic ablation of ULK4 in animal models leads to less efficient SHH pathway induction? It also remains to be resolved how ULK3 and ULK4 act in distinct or common manners to promote SHH signalling. Another remaining question is, whether cell type- and tissue-specific features exist, that play a role in ULK3- versus ULK4-dependent SHH pathway modulation. In particular for the studies on ciliary tip localization of factors, relevant for SHH pathway transduction, a higher temporal resolution will be needed in the future as well as a deeper insight into tissue/ cell type-specific mechanisms. These caveats, mentioned here, don't have to be addressed in new experiments for the revision of this manuscript but could be discussed.

We agree with the reviewer that it would be important to investigate in the future the *in vivo* function Ulk4 in Shh signaling, the relationship between Ulk3 and Ulk4/Stk36, and possible cell type/tissue specificity of these two kinase systems. This will need the generation of single and double knockout mice and examine Hh related phenotypes in different tissues and developmental stages. The precise mechanism by which Ulk4 and Stk36 are translocated to the ciliary tip is also an important and unsolved issue. We include several paragraphs in the “discussion” section to address these outstanding questions for future study.

**Reviewer #2 (Public Review):**
The authors provide solid molecular and cellular evidence that ULK4 and STK36 not only interact, but that STK36 is targeted (transported?) to the cilium by ULK4. Their data helps generate a model for ULK4 acting as a scaffold for both STK36 and its substrate, Gli2, which appear to co-localise through mutual binding to ULK4. This makes sense, given the proposed role of most pseuodkinases as non-catalytic signaling hubs. There is also an important mechanistic analysis performed, in which ULK4 phosphorylation in an acidic consensus by STK36 is demonstrated using IP'd STK36 or an inactive 'AA' mutant, which suggests this phosphorylation is direct.The major strength of the study is the well-executed combination of logical approaches taken, including expression of various deletion and mutation constructs and the careful (but not always quantified in immunoblot) effects of depleting and adding back various components in the context of both STK36 and ULK3, which broadens the potential impact of the work. The biochemical analysis of ULK4 phosphorylation appears to be solid, and the mutational study at a particular pair of phosphorylation sites upstream of an acidic residue (notably T2023) is further strong evidence of a functional interaction between ULK4/STK36. The possibility that ULK4 requires ATP binding for these mechanisms is not approached, though would provide significant insight: for example it would be useful to ask if Lys39 in ULK4 is involved in any of these processes, because this residue is likely important for shaping the ULK4 substrate-binding site as a consequence of ATP binding; this was originally shown in PMID 24107129 and discussed more recently in PMID: 33147475 in the context of the large amount of ULK4 proteomics data released.

The reviewer raised an interesting question of whether ATP binding to the pseudokinase domain of Ulk4 might be required for its function, i.e., by regulating the interaction with its binding partner. In a recent study (Preuss et al. 2020;PMID: 33147475), the critical Lys39 for ATP binding was converted to Arg (KR mutation); however, unlike in most kinases the KR mutation affect ATP binding, the K39R mutation in the Ulk4 pseudokinase did not affect ATP binding although it slightly increased ADP binding (PMID: 33147475). Another mutation made by Preuss et al(PMID: 33147475), N239L, affected protein stability, making it impossible to determine whether this mutation affect ATP binding. Therefore, in the absence of clear approach to perturb ATP binding without affecting the overall structure of Ulk4, it would be challenging to address whether ATP binding regulates the ability of Ulk4 to bind its substrates.Nevertheless, we discuss the possibility that ATP binding might regulate Ulk4/Stk36 interaction and Shh signaling.

The discussion is excellent, and raises numerous important future work in terms of potential transportation mechanisms of this complex. It also explains why the ULK4 pseudokinase domain is linked to an extended C-terminal region. Does AF2 predict any structural motifs in this region that might support binding to Gli2?

The extended C-terminal domain of Ulk4 contains Arm/HEAT repeats (protein-protein interacting domain), which are predicted by AF2 to form alpha helixes.

A weakness in the study, which is most evident in Figure 1, where Ulk4 siRNA is performed in the NIH3T3 model (and effects on Shh targets and Gli2 phosphorylation assessed), is that we do not know if ULK4 protein is originally present in these cells in order to actually be depleted. Also, we are not informed if the ULK4 siRNA has an effect on the 'rescue' by HA-ULK4; perhaps the HA-ULK4 plasmid is RNAi resistant, or if not, this explains why phosphorylation of Gli2 never reaches zero? Given the important findings of this study, it would be useful for the authors to comment on this, and perhaps discuss if they have tried to evaluate endogenous levels of ULK4 (and Stk36) in these cells using antibody-based approaches, ideally in the presence and absence of Shh. The authors note early on the large number of binding partners identified for ULK4, and siRNA may unwittingly deplete some other proteins that could also be involved in ULK4 transport/stability in their cellular model.

Due to the lack of reliable Ulk4 and Stk36 antibodies, we were unable to confirm knockdown efficiency by western blot analysis. Therefore, we relied on the measure Ulk4 and STk36 mRNA expression by RT-qPCR to estimate the knockdown efficiency (Fig 1- figure supplement 1). We used mouse Ulk4 shRNA to carry out the knockdown experiments in NIH3T3 and MEF cells while the human version of Ulk4 (hUlk4) was used for the rescue experiments (Fig 1- figure supplement 2; Fig. 8). We have confirmed that the mUlk4 shRNA targeting sequence is not conserved in hUlk4; therefore, the hULK4 construct is RNAi resistant. The rescue experiments strongly argue that the effect of Ulk4 RNAi on Shh signaling is due to loss of endogenous Ulk4. This argument is further strengthened by the observations that mutations that affected Ulk4 and Stk36 ciliary tip localization also affected Shh signaling such as Gli2 phosphorylation and Ptch1/Gli expression (Fig. 8).

The sequence of ULK4 siRNAs is not included in the materials and methods as far as I can see.

We have added the mouse Ulk4 RNAi target sequence in the revised version.

**Reviewer #3 (Public Review):**
In this manuscript, Zhou et al. demonstrate that the pseudokinase ULK4 has an important role in Hedgehog signaling by scaffolding the active kinase Stk36 and the transcription factor Gli2, enabling Gli2 to be phosphorylated and activated.Through nice biochemistry experiments, they show convincingly that the N-terminal pseudokinase domain of ULK4 binds Stk36 and the C-terminal Heat repeats bind Gli2.Lastly, they show that upon Sonic Hedgehog signaling, ULK4 localizes to the cilia and is needed to localize Stk36 and Gli2 for proper activation.This manuscript is very solid and methodically shows the role of ULK4 and STK36 throughout the whole paper, with well controlled experiments. The phosphomimetic and incapable mutations are very convincing as well. I think this manuscript is strong and stands as is, and there is no need for additional experiments.Overall, the strengths are the rigor of the methods, and the convincing case they bring for the formation of the ULK4-Gli2-Stk36 complex. There are no weaknesses noted. I think a little additional context for what is being observed in the immunofluorescence might benefit readers who are not familiar with these cell types and structures.

We thank this reviewer for the positive comments.

Recommendations For the Authors
**Reviewer #1 (Recommendations For The Authors):**
This elegant study has been thoroughly and thoughtfully designed and the dataset is solid. The biochemistry results are overall very convincing. Some data lack quantification and there needs to be more information on data analyses and statistics. The following suggestions and comments aim at strengthening the manuscript.1. Please provide quantification normalized to input for IP experiments (Figures 1 E - F; Figure 8 C). More information on data analyses and statistics should be provided and included as information in the figure legends.

Thanks for the suggestions, we have done the quantification and statistics analyses for Figures 1E-G and Figure8 C as requested.

1. Did the authors investigate whether overexpressing hULK4 in the control NIH3T3 cells leads to an increase in pS230/232 (related to Figure 1E)? This would nicely support the notion of a promoting effect of ULK4 on GLI2 phosphorylation.

We did not. We speculated that overexpressing hULK4 may not significantly promote GLI2 phosphorylation because Ulk4 is a pseudokinase and endogenous Stk36 (the kinase partner of Ulk4) is limited.

1. The CO-IP experiments to show GLI2 activation were performed in NIH3T3 cells, whereas HEK293 cells were used for the experiments shown in Figure 2. Is there a specific reason for switching between cell lines also for experiments shown in Figures 3 C- I? Did the authors repeat some of the key experiments in both cell lines?

In mammalian cells, Shh-induced activation of GLI2 depends on primary cilia (Han et al., 2019). NIH3T3 cells form the primary cilia but HEK293T cells do not. Therefore, we used NIH3T3 cells to examine the processes that are regulated by the Shh treatment assay (e.g., the Shh-induced phosphorylation of GLI2 and STK36). The HEK293 cells were used to map binding domain between ULK4 and STK36/GLI2/SUFU due to the high transfection efficiency.

1. In Figure 2 D-E the authors nicely showed that hUlk4N-HA interacted with CFP-Stk36 but not with Myc-Gli2/Fg-Sufu whereas hUlk4C-HA formed a complex with Myc-Gli2/Fg-Sufu but not with CFP-Stk36. In Figure 4E the authors showed in their Co-IP experiments that Fg-Stk36 and Myc-Gli2 form a complex independent of SHH treatment. Did the authors see some pull down of Stk36, still in complex with Gli2, using hUlk4C IP and pull down of Gli2, still in complex with Stk36, using hUlk4N IP?

We did not test that. As we have shown in Figures 4A and 4E, knockdown of endogenous ULK4 nearly abolished the interaction between Myc-GLi2 and Fg-Stk36, suggesting that Ulk4 is the major scaffold to bring Skt36 and Gli2 together, and that there is little if any direct interaction between GLi2 and Stk36.

1. Another method to verify hULK4-Stk36-Gli2 complex formation (Figure 4) would be helpful. For example, proximity ligation assays, tripartite split GFP assays, or colocalization based on expansion STED immunofluorescence microscopy could be performed to temporally and spatially resolve localization of Ulk4, Stk36 and Gli2 upon SHH stimulation in the primary cilium

Thanks for the suggestions. We think that our current study using biochemical and cell biology approaches have provide sufficient evidence that Ulk4, Stk36 and Gli2 form complexes. We will keep in mind of those more sophisticated methods in our future endeavors.

1. Please provide more representative images of Ulk4, Stk36 and Gli2 localization in NIH3T3 cells or lower magnification overview images showing more than one cell (Figure 5).

We have provided more representative images in Figure 5- figure supplement 1A-F of the revised manuscript.

1. Confirmation of the results shown in Figure 5 in a second cell line would strengthen the data.

We have confirmed the results in MEFs (see Figure 5- figure supplement 1G-J)

1. Did the authors add immunofluorescence for tubulin as a ciliary base marker to ensure correct assignment of ciliary tip versus ciliary base localization for quantification experiments (Figures 5 - 8)?

It has been well documented that GLi2 is accumulated at the ciliary tip in respond to Shh treatment; therefore, we used Gli2 as a marker for ciliary tip where both Ulk4 and Stk36 were also accumulated. γ tubulin staining could be another marker to assign the ciliary tip vs base; however, the antibody combination we have did not allow us to simultaneously stain γ tubulin and acetylated tubulin (Ac-Tub).

1. SMO localization as a further readout of SHH pathway activation might be considered to be added for some of the key results (e.g., Figure 6). Is SMO trafficking affected after depletion or overexpression of ULK4?

Due to the lack of a workable antibody to detect endogenous Smo in our hands, we did not determine whether the trafficking of SMO is affected after depletion or overexpression of ULK4. However, we noticed that a recent study reported that the SHH-induced ciliary SMO accumulation was impaired in Ulk4 siRNA treated cells (Mecklenburg et al. 2021). We include this information and its implication in the discussion section

1. Do the authors see ULK4 only at the ciliary tip after SHH stimulation or is there also a dynamic time-dependent localization along the ciliary shaft? The image in Figure 6E (dKO + Stk36 WT) seems to show ULK4 also in the shaft.

Unlike Smo that is evenly distributed alone the axoneme of primary cilia, ULK4 is mainly accumulated at ciliary tips upon Shh stimulation. Ulk4 is also located at low levels outside the cilia and sometimes in the ciliary shaft during its transit to the ciliary tip (e.g., see Figure 5- figure supplement 1F1-2; J1-2).

1. Is the immunofluorescence signal for Ulk4 significantly reduced after shRNA treatment to deplete Ulk4 (Figure 6A)?

We constructed a cell line that stably expressed ULK4 shRNA. The knockdown efficiency was determined by measuring Ulk4 mRNA expression (Fig 1_figure supplement 1). Because we were unable to obtain a reliable ULK4 antibody for immunostaining, we did not examine by whether ULK4 signal was depleted by Ulk4 shRNA.

1. The labelled ciliary tip resembles in some cases images seen for ciliary abscission. The authors could use membrane/ciliary membrane markers to ensure "intraciliary" localization of the investigated factors.

Thanks for the suggestion. We will try that in our future experiments.

1. How many replicates were used in the three independent quantitative RT-PCR experiments (Figure 1 A-D)?

We used 3 replicates in each independent quantitative RT-PCR assay.

1. Please provide p values or statement on no significance for the comparison between Ulk3 single and Ulk3/Ulk4 double knockdown (Figure 1C) and between Stk36 single and Stk36/Ulk4 double knockdown (Figure 1D; Fig1_Figure Supplement 2).

Thanks for the suggestion, we have added the p value or “ns” as asked.

1. Figure legends in general are a bit short could have some more detailed information.

Thank you for the suggestion, we have revised the Figure legends as asked.

1. What do the asterisks present in Figure 4 C-D?

Thanks for the suggestion. The asterisks in Figure 4C-D indicated the full length STK36 and truncated form STK36N and STK36C fragments. We that included this information in the figure legend.

1. The authors state that a previous study described ULK4 as a genetic modifier for holoprosencephaly and that this raised the possibility that ULK4 may participate in HH signal transduction. Primary ciliary localization of ULK4 in mouse neuronal tissue and SHH pathway modulation by ULK4 in cell culture have been shown by Mecklenburg et al. 2021 before. Maybe the authors could rephrase their introduction and discussion accordingly.

Thanks for the suggestion, we have changed the introduction and discussion accordingly.

1. Overexpression studies in heterologous systems using tagged proteins can potentially have an influence on their subcellular localization and function. Please discuss this caveat.

We have mentioned this caveat in the “discussion” section of the revised manuscript. However, we have tried to express the transgene at low levels using the lentiviral vector containing a weak promoter to ensure that the exogenously expressed proteins are still regulated by Hh signaling. We have also confirmed that the tagged Ulk4 and Stk36 can rescue the loss of endogenous genes.

1. More details in the Methods section should be provided on the SHH induction in NIH3T3 cells, HEK293 cells and MEFs.

We have revised the methods section on Shh induction.

1. ULK4 is known to have at least three isoforms that exhibit varying abundance across developmental stages in mice and humans (Lang et al., 2014) (DOI:10.1242/jcs.137604). Can the authors speculate on potential common and distinct functions of the different ULK4 isoforms on SHH pathway modulation based on their present results?

It is interesting that Ulk4 has multiple isoforms in both mouse and human. Several short isoforms in both mouse and human lack the pseudokinase domain while one short isoform in mouse lacks the C-terminal region essential for Ulk4 ciliary tip localization. We speculate that the C-terminally deleted isoform may not have a function in the Shh pathway based on our results shown in Fig. 7 and 8 but might still have functions in other cellular processes.

**Reviewer #2 (Recommendations For The Authors):**
The paper is well written, and clear throughout, with excellent (up-to-date) citations to the field.

We thank reviewer #2 for the positive comments.

**Reviewer #3 (Recommendations For The Authors):**
My only quibble is that the immunofluorescence images are a little confusing, especially to people outside of the field. Please include an image of the whole field and improve the captions. Is that a single cell for each cilia? Why are there so few cilia? The DAPI makes it seem like What are we looking at? Are those multiple nuclei in Figure 6? They seem a little small if that's the 5 uM scale bar

We provide uncropped images of Figure 5E to show the entire cells (below). We have added some context to improve the captions. Most of the mammalian cells such as MEF and NIH3T3 cells contain a single primary cilium; however, mutilated cells do exist. The DAPI staining indicated the nuclei. The cells shown in Figure 6 have single nucleus (the scale should be 2 µM). Due to the unevenness of DAPI signals in the nuclei, only the strong signals (puncta) were shown for individual nuclei.

**Author response image 1. sa4fig1:** 

One small typo: GLL2 instead of GLI2 on line 363

Thanks, we have corrected this spelling mistake.